# FLAME:
# REDUCING COMPUTATION IN FEDERATED LEARNING VIA SAMPLE-ADAPTIVE MULTI-EXIT TRAINING

## ABSTRACT

Federated learning (FL) enables a group of clients to collaboratively train a global machine learning model without sharing raw data. It is particularly suited to Internet-of-Things and similar environments involving small, heterogeneous devices. However, these clients often lack the computational resources needed to train the full global model locally, as the FL pipeline conventionally expects. Prior work addresses this challenge by assigning smaller sub-networks to resource-constrained clients, but such approaches have a key limitation: they do not adapt computational effort based on the needs of individual input samples. In this work, we introduce Federated Learning with sample-Adaptive Multi-Exiting (FLAME), the first method to incorporate sample-adaptive early exiting into local training for efficient FL. FLAME allows each training sample to exit at the earliest layer at which the model can confidently predict the sample's output, which improves efficiency without sacrificing accuracy. We show that this use of sample-adaptiveness leads to better AUC than existing solutions because instead of uniformly saving computation across all samples, it strategically saves it on easier samples and preserves it for harder ones. Our empirical results demonstrate FLAME's ability to reduce per-client computation by up to 50% while maintaining or even improving model accuracy, and to outperform existing solutions in practical settings. We also show how FLAME's success stems from FL's collaborative nature and propose two optimizations that further enhance its efficiency and performance. Overall, this work introduces the novel concept of training-time sample-adaptiveness in the FL domain, which opens new avenues for improving the utilization of heterogeneous clients and for enhancing the FL paradigm.

## 1 INTRODUCTION

The cost of training deep learning systems is rising rapidly, with recent models like GPT-4 and Gemini Ultra requiring 10B–100B petaFLOPs (Maslej et al., 2024). In federated learning (FL), this cost is distributed across many clients that train local models on their own data, while a central server aggregates updates into a global model. This enables learning from large, diverse datasets without sharing raw data, e.g., in wearable health applications where FL supports collaborative disease detection while preserving privacy. A central challenge is that clients are often resource-constrained yet still expected to train the full architecture. While storage can be a factor, computation is the primary bottleneck. Training FLOPs are growing exponentially (Amodei & Hernandez, 2018; AI Index Steering Committee, 2025). However, memory demands have increased more slowly or even declined in recent models (Hoffmann et al., 2022). For instance, Hoffmann et al. (2022) show that their Chinchilla model outperforms the SoTA Gopher model that is 4x larger than Chinchilla but has 4x less training data. This result demonstrates the fact that scaling training data is often more beneficial than increasing model size, and consequently, that memory is becoming a less dominant constraint than compute.

We introduce Federated Learning with sample-Adaptive Multi-Exiting (FLAME), a flexible and efficient scheme that reduces local computation by adapting it to input difficulty. FLAME builds on two insights. First, many samples do not require full network depth for accurate predictions, motivating multi-exit models (MEMs) that allow early exits during inference (Kaya et al., 2019).

While MEMs have been used for inference, applying them to training is underexplored and raises a key concern: can the global model still converge if many samples exit early and deeper layers receive fewer updates? Our experiments, supported by theory, show that FLAME maintains convergence in practice. Second, input-adaptive training has improved performance in other contexts such as robust optimization and subgroup generalization via reweighting and resampling (Sagawa et al., 2019; Byrd & Lipton, 2019; Cao et al., 2019; Liu et al., 2021; Nam et al., 2020; Sohoni et al., 2020; Namkoong & Duchi, 2017), but has not been used to reduce computation.

Existing FL approaches for resource-limited clients typically assign smaller sub-networks based on device capacity. These methods suffer from overhead in sub-network generation, limited flexibility, and a narrow focus on device constraints rather than the data itself. This is problematic in non-IID settings where sample difficulty varies. Uniform savings can under-compute on harder samples and over-compute on easier ones. FLAME instead adapts to each sample, allocating more resources to difficult examples and fewer to easier ones.

We evaluate FLAME on language tasks and show it reduces training costs by up to 50% while maintaining or improving accuracy and inference efficiency. Through ablation studies, we show that FL's collaborative setup mitigates under-training of deeper layers and that sample-adaptive computation improves AUC. We introduce a batching optimization for further savings and propose three aggregation algorithms tailored to FLAME. Finally, we demonstrate that FLAME performs especially well under realistic non-IID distributions, outperforming prior methods and confirming that its sample-adaptive design is central to its effectiveness.

**Our main contributions are:**

- Introducing FLAME, the first sample-adaptive, multi-exit training framework for FL, which reduces client computation while often improving accuracy and inference efficiency.
- Showing that FLAME remains stable and convergent despite sample-adaptive exits, supported by both empirical results and an $O(1/T)$ convergence proof.
- Conducting ablation studies to show (1) collaboration mitigates under-training of later layers and (2) sample-level adaptation improves AUC.
- Proposing a grouped backpropagation strategy that further reduces computation, with experiments guiding its tuning.
- Developing and evaluating three aggregation algorithms tailored to FLAME that improve efficiency and accuracy.
- Providing evidence that, on non-IID client distributions with diverse sample difficulty, FLAME achieves higher AUC than state-of-the-art baselines under matched training FLOPs.

## 2 RELATED WORK

### 2.1 EXISTING SOLUTIONS AND THEIR DRAWBACKS

FLAME addresses resource constraints in FL clients, a challenge previously approached by assigning smaller sub-networks for local training (Ilhan et al., 2023; Wang et al., 2024; Niu et al., 2023; Diao et al., 2021; Mei et al., 2022; Varma et al., 2023; Bouacida et al., 2020; Horvath et al., 2022; Kim et al., 2023; Lee et al., 2024; Liu et al., 2022). FLAME's key distinction is sample-specific adaptation: it dynamically adjusts computation based on input difficulty, allocating more resources to harder samples. This leads to higher AUC scores than non-adaptive methods, which often under-train on difficult samples to save compute.

Other limitations in prior work include high overhead and inflexible sub-network assignment. For example, FedDSE (Wang et al., 2024) and PriSM (Niu et al., 2023) rely on expensive supernet training and SVD analysis, respectively, while FjORD (Horvath et al., 2022) uses costly Optimal Dropping. Several methods also lack strategies for assigning architectures based on client-specific characteristics. HeteroFL (Diao et al., 2021) uses fixed downscaling ratios, and ScaleFL (Ilhan et al., 2023) selects from a limited set of depth-width variants. InclusiveFL (Liu et al., 2022) optimizes for participation and utility but does not tailor architectures to client data. Only AFD (Bouacida et al., 2020) and FLANC (Mei et al., 2022) support evolving sub-networks during training, through dynamic pruning and shared basis construction. FLAME offers similar flexibility at lower cost. Clients can

adjust their computation using a simple patience hyperparameter (Section 4), and the sub-portion of the global model used per sample is selected adaptively during training.

It is important to note that FLAME requires clients to store the full global model, which may seem a drawback compared to methods assuming clients cannot. However, FLAME does save some storage by avoiding activations for all parameters (Appendix K). More importantly, as discussed in the introduction, memory limitations are often not the primary constraint in practice and it is the computational cost that presents the more significant challenge. Therefore, the focus of this work is on the reduction of computational costs as opposed to saving memory/communication costs.

## 2.2 Multi-exit models

FLAME is largely inspired by Multi-Exit Models (MEMs). With MEMs, input samples can 'exit early' and thereby receive a final prediction at earlier layers than the final output layer (Zhou et al., 2020; Kaya et al., 2019; Huang et al., 2018; Xin et al., 2020; Liu et al., 2020). The early exiting happens through internal classifiers (ICs), which are attached to various layers within a MEM. During training, ICs learn to map the layer's hidden state to a prediction (referred to as an internal prediction). During inference, samples can exit through an IC once some exiting criteria is met. FLAME is the first solution to integrate the multi-exit approach into the training process, where computational demands are significantly higher than in inference.

## 3 Experiment set-up

**FL setup** We consider a federated learning (FL) system with multiple clients, each training locally on its own data. A central server maintains the global model and aggregates client updates using FedAvg (McMahan et al., 2023), which computes a weighted average based on client dataset sizes. Since data is evenly split, this reduces to a simple average. We assume a synchronous setting where the server waits for all updates before aggregating and then broadcasts the updated global model. Each client trains using the full architecture.

**FL settings** When using FLAME, clients are assigned patience values $p$ controlling early exits: a sample exits once $p$ successive ICs agree (see Section 4). We denote training-time patience as $p^{tr}$, where smaller $p^{tr}$ reduces computation. Table 1 lists 10 settings with different $p^{tr}$ distributions across 10 clients. We focus mainly on 10-client settings, but also include larger, 25-client experiments in Section 7, which aligns with many FL works (e.g. Blanchard et al. (2017); Lin et al. (2021); Li et al. (2023); Tan et al. (2022); Yu et al. (2022); Xenos & Serpanos (2025)) that focus on 10-20 clients in their empirical evaluations. We restrict $p^{tr} \in [2, 6]$. $p^{tr} = 1$ forces exit at the first IC, while $p^{tr} > 6$ makes nearly all samples exit at the final layer, effectively disabling early exiting. Note that while we report results across all 10 settings, we primarily focus on Setting A because it provides a representative mix of low- and high-patience clients, which best illustrates FLAME's collaborative dynamics.

Table 1: List of $p^{tr}$ values used by clients while using FLAME in various 10-client FL settings.

| Setting | Client $p^{tr}$ values |
|---------|------------------------|
| A | [2,2,3,3,4,4,5,5,6,6] |
| B | [2,2,2,2,2,2,2,2,2,2] |
| C | [2,2,2,2,2,2,2,2,2,6] |
| D | [2,2,2,2,2,2,2,2,6,6] |
| E | [2,2,2,2,2,2,2,6,6,6] |
| F | [2,6,6,6,6,6,6,6,6,6] |
| G | [3,3,3,3,3,3,3,3,3,3] |
| H | [3,3,3,3,3,3,3,3,3,6] |
| I | [3,3,3,3,3,3,3,3,6,6] |
| J | [2,2,2,2,2,6,6,6,6,6] |

**Multi-exit model details** In all experiments, we use a BERT-based MEM with internal classifiers (ICs) at each of the 12 hidden layers. Models are initialized with pre-trained weights from Hugging Face (Wolf et al., 2020), then fine-tuned on downstream tasks (see Section 3), rather than trained from scratch. This approach is standard for BERT-based models, which are known to generalize well, and avoids the high cost of pre-training. Further architecture details are in Appendix A.1.

**Tasks** We evaluate on three GLUE benchmark tasks (Wang et al., 2019): SST-2 (Socher et al., 2013), a binary sentiment classification task; MRPC (Dolan & Brockett, 2005), which predicts semantic equivalence between sentence pairs; and MNLI (Williams et al., 2018), a three-way classification task labeling premise-hypothesis pairs as entailment, contradiction, or neutral. SST-2 and MNLI are trained for 10 rounds, MRPC for 20, each using 10 clients with evenly split data (see Appendix A.3). In Section 7, we also evaluate on Sentiment140 (Sent140) (Caldas et al., 2019), a non-IID benchmark where each client holds tweets from a single user. We use 25 clients with at least 100 training samples each and train for 20 rounds (details in Appendix A.3).

**Evaluation metrics and protocols** Unless otherwise noted, we evaluate with evaluation-time patience $p^{ev} = 4$ (the full patience mechanism is in Section 4). We report AUC and the average exit layer, and we state in each table caption whether this average refers to inference-time exits or training-time exits. We sometimes report AUC stratified by early-exiting vs. late-exiting samples, defined by the exit layer taken during inference with $p^{ev} = 4$: a sample is early if it exits strictly before a task-specific threshold $\tau_{task}$, otherwise late. (The $\tau_{task}$ values are given in Appendix A.5.) We also report "seconds per iteration," denoting the wall-clock time to complete one full training iteration over the dataset (forward + backward). Details on learning hyperparameters are in Appendix A.2.

## 4   FLAME - USING MULTI-EXIT TRAINING FOR MORE EFFICIENT FL

In this section, we introduce Federated Learning with sample-Adaptive Multi-Exiting (FLAME)[1] (Figure 1). FLAME mirrors inference-time execution of MEMs, but is applied during training. We assume the model includes internal classifiers (ICs) at hidden layers. The ICs amount to a negligible increase in parameter count (a 0.017-0.025% increase, as detailed in Appendix A.1) and hence in storage and communication costs, as well as a negligible increase in FLOPs (see Appendix B for details). During training, each sample passes through the network and receives classifications from each IC. The forward pass stops once a patience-based criterion is met: the sample exits when $p^{tr}$ successive ICs agree. Backpropagation then starts from that layer, updating only earlier parameters. If no early exit occurs, then full backpropagation proceeds. At evaluation, the same mechanism can apply with patience $p^{ev}$. In practice, clients may also easily adjust their $p^{tr}$ value from round to round to reflect their compute budget (e.g. decreasing it if resources are tight or increasing it when capacity is available) so they can adapt to fluctuating resources without requiring any extra work. This initial version of FLAME assumes single-sample batches during training (SGD), but Section 6.1 introduces a mini-batch adaptation. Appendix C provides pseudocode and Appendix D proves FLAME converges at rate $O(1/T)$ (similar to FedAvg).

Note that, in practice, clients can easily adjust the $p^{tr}$ value they use at every sound according to their current compute budget and use. If their use is over budget, they can decrement $p^{tr}$, and if under, they can increment $p^{tr}$. This lets clients adapt when resources fluctuate, without requiring extra labels or added passes. Therefore, even though we do not have any concrete instructions or heuristics that clients can use for choosing patience values, they can simply start with some modest guess at a patience value and make adjustments during training as they see fit.

Table 2 shows AUC scores and average exit layers for global models trained with FLAME across settings (Table 1). For comparison, we include baselines that do not allow early exiting during training (although, like all other settings, they do allow inference-time early exiting). A consistent pattern emerges: training with multi-exits leads to earlier inference exits, so training-time savings translate to inference efficiency. With SST-2, we see that FLAME improves AUC compared to the baseline. We suspect that this could be due to two reasons. First, with a relatively easy task like SST-2, constantly training the full network may be causing overfitting and consequently subpar generalization to new data. Second, Kaya et al. (2019) introduced early exiting to inference in order

---

[1]A link to the code will be provided in the non-anonymized version of the paper

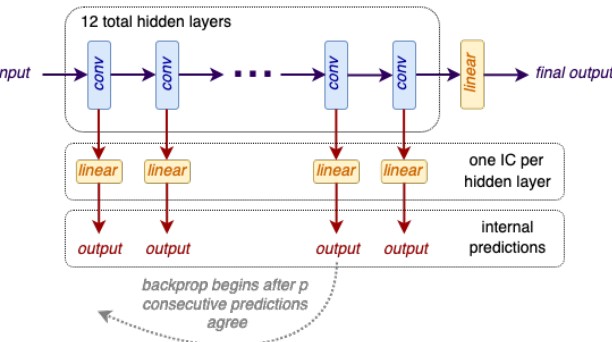

Figure 1: Overview of FLAME. Each layer has an internal classifier (IC). When $p^{tr}$ consecutive IC predictions agree, the sample exits, and backpropagation begins from that layer, skipping later ones.

to evade an overthinking problem that causes a model to incorrectly predict samples if it performs excessive computation on them. Therefore, it's possible that FLAME-trained models, in their ability to cause earlier exiting during inference (compared to the baseline), lead to even less overthinking and hence better evaluation AUC. FLAME maintains baseline AUC on MNLI (a .002 drop is negligible). On MRPC, FLAME slightly reduces AUC except in Setting F, where it improves. This likely reflects the task's difficulty and small size (3069 samples total, only 306 per client), which makes optimization harder under FLAME.

Table 3 compares per-iteration time for SST-2 with and without using FLAME. These values are from the final training round, when client differences are most pronounced. As training progresses, the model increasingly predicts samples earlier, especially for low-patience clients. Appendix Figure 4 shows this trend for MRPC. Cost savings on SST-2 range from 19.34% to 48.32% (Table 3). For MRPC and MNLI, we report 22.18–46.43% and 1.70–28.06% savings in Appendix Table 12.

Table 2: Evaluation AUC score and average exit taken during inference for FLAME, across the settings in Table 1, compared against a baseline that does not allow any early exiting during training. All settings, including the baseline, allow early inference-time early exiting with patience $p^{ev} = 4$.

| **SST-2** | Baseline | A | B | C | D | E | F | G | H | I |
|---|---|---|---|---|---|---|---|---|---|---|
| AUC | 0.83 | 0.98 | 0.88 | 0.89 | 0.88 | 0.89 | 0.92 | 0.92 | 0.91 | 0.91 |
| Avg. exit | 11.44 | 7.83 | 7.13 | 7.37 | 7.28 | 7.67 | 9.97 | 9.41 | 9.64 | 8.93 |

| **MRPC** | Baseline | A | B | C | D | F | J |
|---|---|---|---|---|---|---|---|
| AUC | 0.84 | 0.78 | 0.72 | 0.77 | 0.77 | 0.85 | 0.83 |
| Avg. exit | 11.97 | 6.77 | 11.96 | 10.83 | 8.94 | 11.73 | 11.97 |

| **MNLI** | Baseline | A | B | C |
|---|---|---|---|---|
| AUC | 0.81 | 0.81 | 0.78 | 0.79 |
| Avg. exit | 11.73 | 9.97 | 8.85 | 9.02 |

# 5 DIGGING DEEPER: WHY DOES FLAME WORK?

This section uses ablation studies to highlight two key factors behind FLAME's success: FL's collaborative nature and the multi-exit mechanism's sample-adaptiveness.

## 5.1 ABLATING THE COLLABORATION

FL's collaboration shapes global parameters. Table 4 shows that this collaboration is essential for accurate FLAME training. We trained four centralized models on SST-2 and MRPC with 1/10th of

Table 3: Seconds per iteration on SST-2 for clients using FLAME with various $p^{tr}$ values in Setting A, compared to a baseline without FLAME (no early exits). MRPC and MNLI results are in Appendix E.

| $p^{tr}$ | Average seconds/iteration | % change from baseline |
|---|---|---|
| Baseline | 199.11 | – |
| 2 | 102.90 | -48.32% |
| 3 | 120.09 | -39.28% |
| 4 | 134.41 | -32.50% |
| 5 | 147.43 | -25.96% |
| 6 | 160.60 | -19.34% |

the data. One model served as a baseline with no early exiting, while the others used multi-exiting with $p^{tr} = 2$, 3, or 4. We separately evaluate early- and late-exiting samples, defined by the exit layer taken during inference with $p^{ev} = 4$ (see Appendix A.5 for task-specific thresholds). Baseline models showed relatively small AUC gaps between early and late samples: 11.39% for SST-2 and 15.49% for MRPC. In contrast, FLAME-trained models showed much larger gaps, especially with smaller $p^{tr}$ values. For instance, with $p^{tr} = 4$, the gap grows to 20.27% (SST-2) and 25.37% (MRPC). With $p^{tr} = 2$, it reaches 32.35% and 43.10%. This suggests that centralized FLAME sacrifices AUC for late-exiting samples. Figure 2 highlights the cause. Unlike in models without early exiting, when training with early exits (e.g., $p^{tr} = 4$), early-layer parameters vary much more than later-layer ones. This is because many samples exit early and backpropagate only through initial layers, leaving deeper layers under-optimized. As a result, inference suffers due to poorly trained late-layer parameters. This pattern explains the low AUC in Table 2, Setting B, where all clients use $p^{tr} = 2$. Without clients training later layers, the global model performs poorly. However, in Settings A, C, D, E, and F, where at least some clients use higher $p^{tr}$ values, overall AUC remains higher. These higher-patience clients help compensate for the under-training by low-patience clients. Thus, FLAME performs best when not all clients maximize cost savings. Settings C–F show that even one higher-patience client can significantly offset the learning loss from others.

Table 4: Evaluation AUC scores for a centralized SST-2 model trained with MET at various $p^{tr}$ values. Scores are shown separately for early- and late-exiting samples, defined by the exit layer taken during inference with $p^{ev} = 4$ (thresholds are given in Appendix A.5). A no-MET baseline is included. MRPC results are in Appendix F.

| Method | Early-exiting AUC | Late-exiting AUC |
|---|---|---|
| Baseline | 0.88 | 0.79 |
| MET, $p^{tr} = 2$ | 0.90 | 0.68 |
| MET, $p^{tr} = 3$ | 0.84 | 0.79 |
| MET, $p^{tr} = 4$ | 0.89 | 0.74 |

## 5.2 ABLATING THE SAMPLE ADAPTIVENESS

A key strength of FLAME is its sample-adaptiveness. Table **??** uses the SST-2 task to highlight the benefit of assigning exit points per training sample rather than using a fixed exit for all samples. (Note that both FLAME and fixed-exit models are evaluated using our standard patience-based early exiting with $p^{ev} = 4$.) For instance, a centralized FLAME model with $p^{tr} = 2$ achieves an AUC of .774 and an average exit layer of 6.403. In comparison, models forced to exit at layers 6 and 7 (mimicking the average 6.403 exit) perform significantly worse with AUCs of .591 and .491. This shows that some samples, presumably the more difficult ones, need to pass through many (if not all) network layers. If we force these samples to exit too early, the model is prevented from learning effective representations. FLAME avoids this limitation by letting easy samples exit early while allowing harder samples to traverse more layers. As a result, FLAME can achieve a similar average computational cost while producing better-optimized models than approaches that enforce the same exit for every sample. More broadly, this suggests that models interpret training samples differently, reinforcing the promise of sample-adaptive strategies across other ML workflows. One limitation of

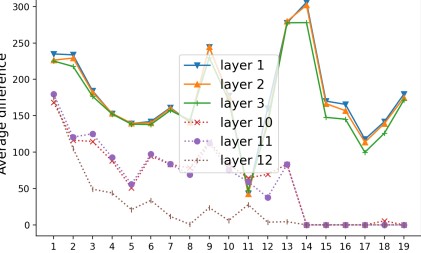
(a) A client using FLAME with $p^{tr} = 2$.

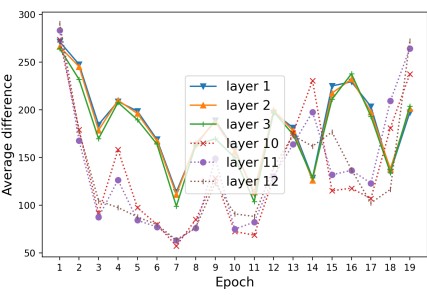
(b) A client using FLAME with $p^{tr} = 6$.

Figure 2: Average difference in local model parameters before and after a client trains for one local round. Clients are using FLAME with different $p^{tr}$ values to learn the MRPC task in a FL system using Setting A. We plot differences by layer and across all rounds of training.

this setup is the fact that the FLAME models and fixed-exit models, though trained differently, use the same exact evaluation scheme. Therefore, we also include additional results for a similar experiment, but now using MRPC instead of SST-2. Here, we still use $p^{ev} = 4$, but for the fixed-exit-trained models, during evaluation, we do not allow samples to exit any later than last layer that samples could reach during training. These new results are in Table 6 and they again demonstrate the advantage of samples-specific exiting. Across comparable compute budgets, FLAME models achieve higher AUC scores than fixed-exit baselines.

Table 5: Evaluation AUC, average exit layer (across all training rounds), and seconds per iteration (last round) for SST-2 models. Some use patience-based early exiting with various $p^{tr}$ values and others enforce fixed exit layers for all samples. For all models, patience-based early exiting is used during evaluation with $p^{ev} = 4$.

| Exit strategy | AUC | Average exit taken during training | Seconds/iteration |
| --- | --- | --- | --- |
| FLAME, $p^{tr} = 2$ | 0.774 | 6.403 | 115.24 |
| FLAME, $p^{tr} = 3$ | 0.877 | 9.984 | 164.36 |
| FLAME, $p^{tr} = 4$ | 0.878 | 11.555 | 186.27 |
| FLAME, $p^{tr} = 5$ | 0.870 | 11.961 | 191.74 |
| FLAME, $p^{tr} = 6$ | 0.881 | 11.999 | 192.66 |
| Exit layer 6 | 0.591 | 6 | 113.59 |
| Exit layer 7 | 0.491 | 7 | 126.49 |
| Exit layer 8 | 0.482 | 8 | 140.80 |
| Exit layer 9 | 0.483 | 9 | 156.25 |
| Exit layer 10 | 0.509 | 10 | 170.57 |
| Exit layer 11 | 0.846 | 11 | 197.52 |

# 6 EXPLORING FLAME EXTENSIONS

## 6.1 ENABLING LARGER BATCH SIZES

A drawback of FLAME is its reliance on batch size 1, which requires stochastic gradient descent and is often less efficient than mini-batch approaches. To address this, we propose *grouped backpropagation*, which retains FLAME's per-sample forward pass while enabling batched backpropagation. After $b$ samples complete forward passes, they are grouped by exit layer. With 12 ICs, this yields up to 12 groups, though most are empty. Figure 3 shows this clustering effect for SST-2. For each non-empty group, losses are averaged and a single backpropagation step is performed for the averaged loss. Table 7 shows that this reduces training cost on SST-2 in Setting A. Per-client results are in Appendix Table 14. While AUC drops slightly with grouping, it remains above the no-FLAME baseline, so

Table 6: Evaluation AUC, average exit layer (across all training rounds), and average FLOPS per sample per one full iteration (calculated using the formulas explained in Appendix B) for MRPC models. Some use patience-based early exiting with various $p^{tr}$ values and others enforce fixed exit layers for all samples. All models use patience-based early exiting during evaluation ($p^{ev} = 4$). The fixed-exit models do not allow exits after the layer that training stopped at.

| Exit strategy | AUC | Average exit during training | Average MFLOPS/sample/iteration |
|---|---|---|---|
| FLAME, $p^{tr} = 2$ | 0.765 | 6.774 | 81859 |
| FLAME, $p^{tr} = 3$ | 0.713 | 10.700 | 129302 |
| FLAME, $p^{tr} = 4$ | 0.738 | 10.854 | 131163 |
| FLAME, $p^{tr} = 5$ | 0.755 | 11.752 | 142015 |
| FLAME, $p^{tr} = 6$ | 0.760 | 11.906 | 143876 |
| Exit layer 6 | 0.691 | 6 | 72506 |
| Exit layer 7 | 0.708 | 7 | 84590 |
| Exit layer 8 | 0.712 | 8 | 96674 |
| Exit layer 9 | 0.727 | 9 | 108759 |
| Exit layer 10 | 0.721 | 10 | 120843 |
| Exit layer 11 | 0.711 | 11 | 132927 |

we view this trade-off as minor. Table 7 also compares grouping strategies. "Full group" places all samples in one group. "Random" assigns samples to one of 12 groups at random. "Binary" splits samples by exit layers 1–6 vs. 7–12. "Distant pairing" forms six groups using (1,7), (2,8), …, (6,12). "Close pairing" uses (1,2), (3,4), …, (11,12). Our proposed strategy consistently achieves the highest AUC, especially with $b = 32$ and $b = 64$. We believe this is because our method ensures all samples in a group compute gradients for the same parameters. Other strategies average gradients across samples that may not have reached all layers, introducing zeros and distorting updates. Our method avoids this by averaging either valid gradients or zeros exclusively, ensuring consistent updates or none at all. Pseudocode for grouped backpropagation is provided in Algorithm 2 of Appendix C.

Table 7: Evaluation AUC on SST-2 for FLAME with different grouping strategies and $b$ values in Setting A, compared against (1) no FLAME and (2) FLAME without grouping. We also report average seconds per iteration across 10 clients (per-client results in Table 14, Appendix H).

| Grouping strategy | $b$ | AUC | Average seconds/iteration | % change in seconds/iteration |
|---|---|---|---|---|
| No FLAME | 1 | 0.830 | 199.11 | - |
| Standard FLAME | 1 | 0.981 | 132.98 | -33.21% |
| Proposed grouping | 16 | 0.891 | 92.84 | -53.37% |
| Proposed grouping | 32 | **0.913** | 115.75 | -41.87% |
| Proposed grouping | 64 | 0.912 | 99.71 | -49.92% |
| Proposed grouping | 128 | 0.899 | 107.49 | -46.01% |
| Full group | 8 | 0.896 | 97.56 | -51.00% |
| Full group | 32 | 0.893 | 101.93 | -48.81% |
| Binary grouping | 32 | 0.865 | 87.48 | -56.06% |
| Random grouping | 32 | 0.874 | 90.74 | -54.43% |
| Distant pairing | 32 | 0.890 | 100.57 | -49.49% |
| Close pairing | 32 | 0.843 | 99.39 | -50.08% |

## 6.2 ADAPTING AGGREGATION TO FLAME

We explore three aggregation methods tailored to FLAME, each modifying how client models are weighted. The first, ***patience-conscious aggregation***, weights each client $m$'s model $W_m$ by its training-time patience value $p_m^{\text{tr}}$, based on the idea that higher $p^{\text{tr}}$ leads to more optimized parameters. The aggregated model is computed as $W_g = \frac{\sum_{m \in \mathcal{M}} p_m^{\text{tr}} W_m}{\sum_{m \in \mathcal{M}} p_m^{\text{tr}}}$. However, as Figure 5 (Appendix I)

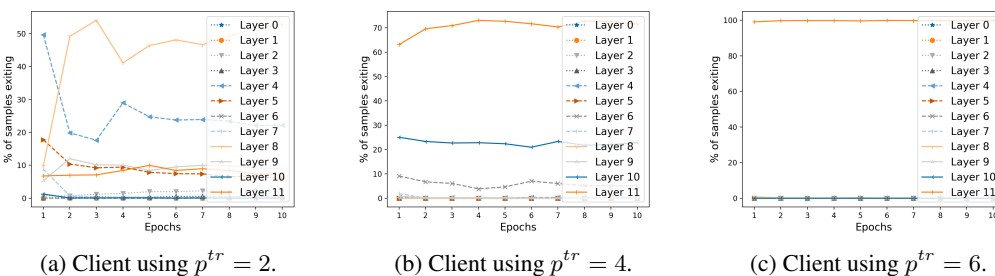

(a) Client using $p^{tr} = 2$.  (b) Client using $p^{tr} = 4$.  (c) Client using $p^{tr} = 6$.

Figure 3: For clients that train with FLAME using different $p^{tr}$ values, we plot the percentage of training samples (out of 4,000 total) that exit at each of the 12 total local model layers. Clients were learning the SST-2 task in Setting A.

shows, this assumption does not always hold: lower-patience clients may exit later. To address this, ***exit-conscious aggregation*** uses each client's average exit layer $e_m$ instead, so the aggregated model is $W_g = \frac{\sum_{m \in \mathcal{M}} e_m W_m}{\sum_{m \in \mathcal{M}} e_m}$. This approach better reflects how deeply each client actually trains its model. Finally, ***exit-based aggregation***, inspired by HeteroFL (Diao et al., 2021), aggregates each layer using only the clients that trained it. Let $W_{m,\ell}$ and $W_{g,\ell}$ denote the parameters of layer $\ell$ in the client and global models, respectively. Then the update is $W_{g,\ell} = \frac{\sum_{m \in \mathcal{M}} \mathbf{1}[e_m \geq \ell] W_{m,\ell}}{\sum_{m \in \mathcal{M}} \mathbf{1}[e_m \geq \ell]}$ for $\ell = 1, \ldots, L$, where $\mathbf{1}[e_m \geq \ell]$ is an indicator that client $m$ reached layer $\ell$. This isolates meaningful updates and avoids penalizing early-layer parameters just because a client did not train deeper layers. Unlike the previous two, which downweight under-trained parameters, exit-based aggregation fully excludes them. While more fine-grained, it still relies on average exits, making it approximate. As shown in Table 8, exit-conscious aggregation generally outperforms patience-conscious, offering slightly better AUCs and shallower exits. Comparing exit-conscious and exit-based, results are mixed. Despite being more targeted, exit-based may be limited by its reliance on average exit layers. In contrast, the mild noise in exit-conscious may help generalization through implicit regularization.

Table 8: Evaluation AUC scores and average exit layers (associated with exits taken during inference with $p^{ev} = 4$) from training with FLAME in Setting A using different aggregation algorithms.

| Task | Aggregation algorithm | AUC | Average exit layer taken during inference |
|------|----------------------|-----|-------------------------------------------|
| SST-2 | Baseline (FedAvg) | 0.891 | 7.828 |
| | Patience-conscious | 0.890 | 8.083 |
| | Exit-conscious | 0.893 | 7.966 |
| | Exit-based | **0.894** | 7.864 |
| MRPC | Baseline (FedAvg) | 0.780 | 6.765 |
| | Patience-conscious | 0.793 | 8.122 |
| | Exit-conscious | 0.801 | 8.042 |
| | Exit-based | **0.804** | 8.167 |
| MNLI | Baseline (FedAvg) | 0.806 | 9.971 |
| | Patience-conscious | 0.811 | 10.396 |
| | Exit-conscious | **0.813** | 10.163 |
| | Exit-based | 0.805 | 9.784 |

## 7 DEMONSTRATING FLAME'S ADVANTAGE OVER PRIOR WORKS

We conduct a larger, real-world experiment to compare FLAME against three representative methods: HeteroFL, ScaleFL, and AFD. These baselines capture the main strategies for reducing client-side computation in FL. HeteroFL reduces channel width with a fixed ratio $r$ per client (Diao et al., 2021). ScaleFL generalizes this idea by selecting from width–depth variants (Ilhan et al., 2023), while AFD

dynamically prunes channels during training to construct client-specific subnetworks (Bouacida et al., 2020). Together, these methods represent static (HeteroFL, ScaleFL) and dynamic (AFD) strategies for subnetwork selection. However, all allocate computation uniformly across a client's data, without adapting to sample difficulty. FLAME instead allows easier samples to exit early while allocating more computation to harder ones. With equal computation, we therefore expect FLAME to use resources more strategically.

All methods are trained on Sent140 with 25 clients over 20 rounds and evaluated on SST-2 (setup details in Appendix A.4). We use $p^{tr} = 3$ for FLAME, yielding a 9.8 average exit layer, which equates to 118.44 GFLOPs per client per training sample. Baseline hyperparameters are tuned for a similar compute budget (see Appendix B): $r = 0.9$ for HeteroFL, $(d = 11, r = 0.9)$ for ScaleFL, and $\delta = 0.2$ for AFD. FLAME uses $p^{ev} = 4$ during evaluation, but since none of the other methods train ICs, they cannot enable inference-time early exiting in the same way so we evaluate the full global model as normal for HeteroFL, ScaleFL, and AFD.[2] Table 9 shows that FLAME achieves the highest overall AUC and outperforms all baselines on both early- and late-exiting samples. HeteroFL performs well on early samples but drops on late ones, likely because uniform width scaling under-computes harder examples. ScaleFL also struggles on late samples despite a competitive early-sample AUC, suggesting that joint width–depth scaling still fails to adapt to per-sample difficulty. AFD, while conceptually dynamic, lags in overall performance under our setup. This could be because fixed-ratio structured pruning from round 1 removes capacity needed for difficult examples and, by reducing parameter overlap across clients, weakens FedAvg aggregation. Overall, FLAME delivers the best AUC under a controlled compute budget, with adaptive allocation leading to clear advantages on both easy and hard examples. In Appendix J, we include similar results for a 50-client setup where again, FLAME achieves superior performance.

Table 9: Comparing FLAME, HeteroFL, ScaleFL, and AFD final global model evaluation AUC scores on the Sent140 task. AUC scores are shown separately for early- and late-exiting samples, defined by the exit layer taken during inference with $p^{ev} = 4$ (thresholds provided in Appendix A.5). We train for 20 rounds with 25 clients that all use the same values for $p^{tr}/r/\delta$. Methods are compute-matched to similar FLOPs (see Appendix B). FLAME enables inference-time early exiting with $p^{ev} = 4$. The other methods are not designed to train ICs so they do not enable any early exiting during evaluation.

| Method | Parameter | AUC | | Average GFLOPs/client |
|---|---|---|---|---|
| | | Early samples | Late samples | |
| FLAME | $p^{tr} = 3$ | **0.8739** | **0.7258** | 118.44 |
| HeteroFL | $r = .9$ | 0.7680 | 0.7207 | 117.41 |
| ScaleFL | $d = 11, r = .9$ | 0.7859 | 0.6393 | 107.63 |
| AFD | $\delta = .2$ | 0.5311 | 0.5573 | 115.96 |

## 8    CONCLUSION

FLAME demonstrates that sample-adaptive multi-exit training can substantially reduce client computation in FL while preserving accuracy, achieving up to a 50% reduction in training time with AUC maintained or improved. Although this paper focuses on saving computational costs, one interesting direction to study further would be to pair FLAME with some additional FL algorithm that focuses more on saving communication or memory costs in order to maximize savings for both communication, memory, and compute. FLAME also opens several additional avenues for future research, for instance in security and privacy. Early exits could enable slowdown-style attacks that delay exits (Hong et al., 2021; Varma et al., 2024; Zhang et al., 2023; Chen et al., 2023; Coalson et al., 2023) or leakage from exit-layer signals (Shokri et al., 2017), but these risks may be mitigated with adversarial training (e.g., Varma et al. (2024) for slowdown robustness) and differential privacy (Abadi et al., 2016). In a wider sense, FLAME highlights sample-adaptiveness as a new paradigm for FL, one that could inspire approaches not only for efficiency but also for fairness, robustness, personalization, and other objectives.

---

[2]It is worth noting that this means there is an unreported benefit present. FLAME is enabling computational cost savings during inference while the other methods are not.

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

## A  ADDITIONAL EXPERIMENTAL SETUP DETAILS

### A.1  MODEL ARCHITECTURE AND INTERNAL CLASSIFIER DETAILS

Our architecture is based on the PABEE model (Zhou et al., 2020), which modifies BERT-base by attaching an internal classifier (IC) at the output of each of its 12 hidden layers. Each IC consists of a linear projection from the hidden state to a softmax layer that outputs class probabilities. All IC parameters are updated during training along with the rest of the model parameters. All models are initialized with pre-trained weights from Hugging Face (Wolf et al., 2020) and then fine-tuned on downstream tasks.

We consider the added storage and communication costs associated with the addition of ICs to be negligible. Since each hidden state has 768-dimensional output, for a task with $y$ output labels, the total number of parameters that the ICs add is $(768 * y + y) * 12$. For $y = 2$, which is the case for SST-2 and MRPC, this equates to 18456 additional parameters. This is an approximately .017% increase from the total 110M parameters that the typical BERT base model has. For MNLI, with $y = 3$, this becomes 27684 parameters, which is a .025% increase.

### A.2  OPTIMIZATION AND LEARNING RATE SCHEDULE

All models are fine-tuned using the AdamW optimizer with $\beta$ values set to (0.9, 0.999) and $\epsilon = 1\mathrm{e}{-8}$. We use a linearly decaying learning rate schedule across training rounds. Training always begins with a learning rate of $2\mathrm{e}{-5}$ and decays uniformly. Specifically, we use the following learning rates.

For 10 rounds of training: $[2\mathrm{e}{-5}, 1.8\mathrm{e}{-5}, 1.6\mathrm{e}{-5}, \ldots, 4\mathrm{e}{-6}, 2\mathrm{e}{-6}]$

For 20 rounds: $[2e{-}5, 1.9e{-}5, 1.8e{-}5, \ldots, 2e{-}6, 1e{-}6]$

This schedule is applied consistently across all clients and tasks.

### A.3 SPLITTING TRAINING DATA ACROSS CLIENTS

For our FL experiments, we split a task's training data such that each client has an equal number of samples associated with each label. In Table 10, we provide the cardinality of the resulting local training datasets. Note that we usually do not allow any clients to share the same sample, but we do allow some overlap with MRPC in Table 8 of Section 6.2 since MRPC has such a small training dataset.

### A.4 ADDITIONAL SENT140 DETAILS

For our experiments in Section 7, we select 25 clients from the Sentiment140 benchmark that have at least 100 training samples. This yields the following per-client training set sizes: 549, 246, 281, 192, 227, 248, 213, 195, 202, 179, 207, 216, 171, 107, 103, 118, 113, 101, 112, 106, 189, 212, 114, 117, and 102.

Table 10: Cardinalities of the local training datasets used by each of 10 FL clients.

| Task | Total # training samples | # samples per client | # per class breakdown |
|---|---|---|---|
| SST-2 | 67349 | 4000 | positive: 2000 negative: 2000 |
| MRPC (no overlap) | 3069 | 306 | equivalent: 101 not equivalent: 205 |
| MRPC (with overlap) | 3069 | 1000 | equivalent: 500 not equivalent: 500 |
| MNLI | 392702 | 1500 | neutral: 5000 entailment: 5000 contradiction: 5000 |

### A.5 DISTINGUISHING TEST SAMPLES AS EARLY- OR LATE-EXITING

In Table 4 of Section 5.1, Table 9 of Section 7, and Table 13 of Appendix G, we report separate AUC scores for early- and late-exiting test samples, defined by the exit layer taken during inference with $p^{ev} = 4$. We determined this early/late distinction by training centralized multi-exit models on each task's full training dataset, using these models to run inference on all test samples while allowing patience-based early exiting with $p^{ev} = 4$, and noting the exit layer taken by each sample. For SST-2, if the sample exited before layer 9, it was considered early-exiting, and it was otherwise considered late-exiting. Out of 872 total test samples, this splitting process resulted in 340 early-exiting samples and 532 late-exiting samples. For MRPC, we similarly split samples on exit layer 8 and end up with 363 early-exiting and 236 late-exiting samples from the 599-sample test dataset. We experimented with various $p^{ev}$ values for inference and different ranges of exit layers to define the early and late split and ultimately selected the values that produced the most balanced divisions.

## B FLOPS COMPUTATIONS AND COMPARISONS

To compare the computational cost across FLAME, HeteroFL, ScaleFL, and AFD, we compute FLOPs for a single training iteration on one sample for one client under various hyperparameters. Table 11 reports total training MFLOPs (forward + backward pass). Each individual row corresponds to using FLAME with average exit depth $d$ to define a target compute budget. Other columns correspond to other methods, and we select hyperparameters for those methods that lead to an amount of FLOPs closest to the row's target compute budget.

We use the following forward-pass formulas, then multiply by 3 to obtain training MFLOPs (backward $\approx 2\times$ forward). A full 12-layer BERT forward costs 48,318.4 MFLOPs, so a full training pass costs 144,955.2 MFLOPs.

$$\text{FLAME (forward):} \quad \left(\frac{d}{12}\right) \cdot 48{,}318.4 \; + \; (1.5711 \cdot d) \tag{1}$$

$$\text{HeteroFL (forward):} \quad r^2 \cdot 48{,}318.4 \tag{2}$$

$$\text{ScaleFL (forward):} \quad \left(\frac{d_s}{12}\right) \cdot \left(r_s^2 \cdot 48{,}318.4\right) \tag{3}$$

$$\text{AFD (forward):} \quad (1 - \delta) \cdot 48{,}318.4 \tag{4}$$

Here, $d$ is the average exit depth (FLAME), $r$ is the width ratio (HeteroFL), $(d_s, r_s)$ are ScaleFL's depth/width settings, and $\delta$ is the dropout ratio for AFD. Note that we implement AFD by selecting a fixed fraction of attention heads and feed-forward neurons per layer. In the formula we use for AFD MFLOPs, we approximate forward MFLOPs as linear in the keep probability $(1 - \delta)$, which slightly overcounts constant terms (e.g., LayerNorm, residual, classifier) that do not scale with $\delta$. In the table, hyperparameters are chosen from a one-decimal grid to be closest to the FLAME budget and shown in parentheses.

Table 11: Total training MFLOPs (forward + backward per sample) for FLAME, HeteroFL, ScaleFL, and AFD. Each row uses FLAME at depth $d$ to define a target compute budget. Parentheses indicate the parameter settings: HeteroFL ($r$), ScaleFL ($d_s, r_s$), and AFD ($\delta$).

| $d$ | FLAME | HeteroFL ($r$) | ScaleFL ($d_s, r_s$) | AFD ($\delta$) |
|---|---|---|---|---|
| 1 | 12,084.31 | 13,045.97 (0.3) | 12,079.60 (4,0.5) | 14,495.52 (0.9) |
| 2 | 24,168.63 | 23,192.83 (0.4) | 23,676.02 (4,0.7) | 28,991.04 (0.8) |
| 3 | 36,252.94 | 36,238.80 (0.5) | 35,514.02 (6,0.7) | 43,486.56 (0.7) |
| 4 | 48,337.26 | 52,183.87 (0.6) | 47,352.03 (8,0.7) | 43,486.56 (0.7) |
| 5 | 60,421.57 | 52,183.87 (0.6) | 61,847.55 (8,0.8) | 57,982.08 (0.6) |
| 6 | 72,505.88 | 71,028.05 (0.7) | 69,578.50 (9,0.8) | 72,477.60 (0.5) |
| 7 | 84,590.19 | 92,771.33 (0.8) | 88,060.28 (9,0.9) | 86,973.12 (0.4) |
| 8 | 96,674.51 | 92,771.33 (0.8) | 97,844.76 (10,0.9) | 101,468.64 (0.3) |
| 9 | 108,758.82 | 117,413.71 (0.9) | 107,629.24 (11,0.9) | 115,964.16 (0.2) |
| 10 | 120,843.14 | 117,413.71 (0.9) | 107,629.24 (11,0.9) | 115,964.16 (0.2) |
| 11 | 132,927.45 | 144,955.20 (1.0) | 132,875.60 (11,1.0) | 130,459.68 (0.1) |
| 12 | 145,011.76 | 144,955.20 (1.0) | 144,955.20 (12,1.0) | 144,955.20 (0.0) |

## C  PSEUDOCODE FOR THE FLAME PIPELINE

## D  THEORETICAL CONVERGENCE GUARANTEE

We very closely model our proof of FLAME's convergence off of that of Federated Partial Model Training (FedPMT) (Wu et al., 2023). With FedPMT, all clients compute the forward pass through the entire model as usual. Backpropagation also starts from the output layer as usual, but it can finish before reaching the shallowest layers. Each client is assigned a parameter value that indicates how many layers are updated during backpropagation. Thus, with FedPMT, like with FLAME, not all layers are consistently updated during training.

The two methods differ in two key aspects. First, with FedPMT, the depth of the network that receives updates is client-specific and fixed throughout training. With FLAME, the utilized network depth is sample-specific and may change across rounds. Second, with FedPMT, the deeper layers are prioritized and the early layers may be skipped due to backpropagation stopping early. With FLAME, the early layers are always updated and the later layers may be skipped due to forward propagation stopping early.

## D.1 Preliminaries

### D.1.1 Surrogate loss function

Before proceeding, we first define $\widetilde{f}_k$, which is the local surrogate loss for a client $k$ that uses FLAME. Since FLAME allows clients to use early exits, not every sample produces gradients for all layers. Therefore, this objective is just the usual training loss, but averaged not only over data samples but also over the randomness of exits. Formally, for a model parameter vector $w$ and a client $k$ with data distribution $\mathcal{D}_k$,

$$\widetilde{f}_k(w) = \mathbb{E}_{(x,y)\sim\mathcal{D}_k}\,\mathbb{E}_{j\sim q(\cdot|x)}\left[\ell_j(w;x,y)\right],$$

where $\ell_j(w;x,y)$ denotes the loss computed at exit $j$ and $q(j \mid x)$ is the exit distribution for input $x$. We can then define the global surrogate objective as the average of this surrogate loss across all clients:

$$\widetilde{F}(w) \;=\; \frac{1}{K}\sum_{k=1}^{K}\widetilde{f}_k(w).$$

### D.1.2 Assumptions

To aid in our proof, we list the following standard assumptions, which are also used in the convergence analyses of FedAvg (Li et al., 2020b) and FedPMT (Wu et al., 2023). The first two assumptions are standard (for example, used in Stich (2019); Li et al. (2020a;b); Wu et al. (2023)). Assumptions 3 and 4 have been used in similar convergence analyses, such as Stich (2019); Li et al. (2020b); Wu et al. (2023); Baek et al. (2021); Zhang et al. (2013); Yu et al. (2018); Stich et al. (2018). We also define a new fifth assumption, which is necessary to ensure that no layer is completely deprived of updates.

**Assumption 1 (smoothness).** *Each client's surrogate loss is L-smooth: for all $w$ and $w'$, $\widetilde{f}_k(w) \leq \widetilde{f}_k(w') + (w - w')^T \nabla \widetilde{f}_k(w') + \frac{L}{2}\|w - w'\|^2$*

**Assumption 2 (strong convexity).** *Each client's surrogate loss is $\mu$-strongly convex: for all $w$ and $w'$, $\widetilde{f}_k(w) \geq \widetilde{f}_k(w') + (w - w')^\mathsf{T}\nabla\widetilde{f}_k(w') + \frac{\mu}{2}\|w - w'\|^2$*

**Assumption 3 (bounded variance).** *Stochastic gradients have bounded variance $\sigma^2$: for all $w$, $\mathbb{E}_{(x,y)\sim\mathcal{D}_k}\left\|\nabla\ell(w;x,y) - \nabla\widetilde{f}_k(w)\right\|^2 \leq \sigma^2$.*

**Assumption 4 (heterogeneity bound).** *The variance across client gradients is bounded by $\zeta^2$: for all $w$, $\frac{1}{K}\sum_{k=1}^{K}\left\|\nabla\widetilde{f}_k(w) - \nabla\widetilde{F}(w)\right\|^2 \leq \zeta^2$*

**Assumption 5 (update probability).** *Each model parameter has a nonzero probability of being updated: $\rho > 0$, where $\rho$ is the minimum probability (taken across all layers) that a layer contributes a gradient update (i.e. a sample doesn't exit before the layer).*

## D.2 Proof

**Proposition 1 (Downhill gradients).** *For each client's surrogate loss, $\widetilde{f}_k$, the assumption of strong convexity (Assumption 2) implies that the stochastic gradient step is a valid descent step (is directed downhill).*

*Formally, for any parameter vector $w$ and optimal parameters $w^*$,*

$$\langle w - w^*, \nabla\widetilde{f}_k(w)\rangle \;\geq\; \widetilde{f}_k(w) - \widetilde{f}_k(w^*) + \frac{\mu}{2}\|w - w^*\|^2.$$

Note that, unlike FedPMT's Proposition 1, we do not introduce an $\epsilon$ term to capture the information loss that results from not consistently updating the full network. In FLAME, the effect of early exits

is already accounted for in the surrogate loss, which averages over all possible exits. The impact of reduced gradient information instead appears later, in Lemmas 1 and 2 and Theorem 1, through a $1/\rho$ factor that reflects the minimum probability that an arbitrary network layer is updated.

**Lemma 1** (**Variance under exits.**). *Here, we adapt Lemma 1 of FedPMT (Wu et al., 2023). Formally, using Assumption 3 and Assumption 5, for global round $t$, the variance of the global surrogate gradient is bounded as*

$$\mathbb{E}\big[\|\nabla\widetilde{F}(w^t) - \nabla\overline{F}(w^t)\|^2\big] \ \leq\ \frac{2\sigma^2}{\rho}.$$

$\nabla\widetilde{F}(w^t)$ *denotes the expected global surrogate gradient, defined as the average of the client-level surrogate gradients $\nabla\widetilde{f}_k(w_k^t)$ (which is itself defined as an expectation over minibatches and exits). $\nabla\overline{F}(w^t)$ is the empirical global surrogate gradient, defined as the average of the sampled client-level surrogate gradients $\nabla\widetilde{f}_k(w_k^t; \xi_k, j)$, where $\xi_k$ is a minibatch sampled from $\mathcal{D}_k$ and $j$ is an exit sampled from $q(\cdot \mid x)$.*

The key difference from this lemma and the corresponding one from FedPMT is the noise term that is multiplied by $2\sigma^2$ to define the variance bound. In FedPMT, the noise term, $|I|\psi$, reflects client-level masking. In FLAME, masking occurs at the sample level so the noise term becomes $1/\rho$, where $\rho$ denotes the minimum probability that any layer is updated. This $1/\rho$ term represents the fact that layers that update less frequently receive fewer gradient contributions, which increases the variance of their estimates relative to layers that are updated more often. When $\rho$ is small, the stochastic gradient for that block is based on less information so the noise must be inversely proportional to $\rho$.

Apart from the modified noise term, the proof for this lemma exactly mirrors that of FedPMT.

**Lemma 2** (**Single-round improvement**). *Under Assumptions 1-5, in the $(t+1)$-th global round, the expected distance between the current global model $w^{t+1}$ and the optimal solution $w^*$ satisfies*

$$\mathbb{E}\|w^{t+1} - w^*\|^2 \ \leq\ (1 - \eta_t\mu)\,\mathbb{E}\|w^t - w^*\|^2 + \eta_t^2\left(8(\tau-1)^2 G^2 \ +\ 2L\zeta^2 \ +\ \frac{2\sigma^2}{\rho}\right),$$

*where $\eta_t$ is the learning rate in round $t$ and $\tau$ is the number of local steps.*

Note that FedPMT used a different term to represent client heterogeneity: $2L\,\eta_t^2\big(|I|\psi + |S| + \varepsilon\big)\Lambda$. This was designed to accommodate FedPMT's client-wise masking design. For FLAME, we harness Assumption 4 and replace this heterogeneity term with the standard $2L\,\zeta^2$ bound (as is used in the FedAvg proof of convergence). The other differences are that FLAME does not require $\varepsilon$ and that it uses $1/\rho$ in place of $|I|\psi$ (as justified earlier in the proof).

**Theorem 1** (**Convergence of FLAME**). *Under Assumptions 1-5, using step size $\eta_t = \frac{2}{\mu(t+\lambda)}$, the convergence of FLAME satisfies*

$$\mathbb{E}\left[\widetilde{F}(w^T) - \widetilde{F}(w^*)\right] \ \leq\ \frac{1}{T+\lambda}\left(\frac{\lambda+1}{2}\Gamma_1 \ +\ \frac{2\widetilde{\Delta}}{\mu^2}\right),$$

*where $\lambda > 0$, $\Gamma_1 \ =\ \mathbb{E}\|w^1 - w^*\|^2$, and $\widetilde{\Delta} \ =\ 8(\tau-1)^2 G^2 \ +\ 2L\zeta^2 \ +\ \frac{2\sigma^2}{\rho}$.*

This result shows that FLAME has the same $\mathcal{O}(1/T)$ convergence rate as FedAvg and FedPMT. The differences lie in the constants. Specifically, FLAME replaces FedPMT's client-wise masking factors $(|I|\psi + |S| + \varepsilon)\Lambda$ with the heterogeneity bound $\zeta^2$, eliminates the $\varepsilon$ term, and incorporates a $1/\rho$ factor to account for sample-specific exits. Apart from these modifications, the proof follows the same telescoping argument as supported in FedPMT (Wu et al., 2023).

# E  SECONDS PER ITERATION FOR MRPC AND MNLI CLIENTS

Here, we include additional results associated with Section 4. In Table 12, we list the seconds per training iteration for clients using FLAME with different $p^{tr}$ values and learning the MRPC and MNLI tasks in Setting A. Compared to a baseline where FLAME is not used, we see 22.18-46.43% and 1.70-28.06% savings with MRPC and MNLI.

# F   PLOTTING COST ACROSS ROUNDS OF TRAINING WITH FLAME

Figure 4 plots the seconds per iteration associated with each of 10 clients that are training using FLAME in a FL system that is learning the MRPC task using Setting A. Clients are using different $p^{tr}$ values with FLAME. This plot illustrates the pattern that we have also noticed with SST-2 and MNLI, and suspect to be a general pattern: clients' time per iteration decreases as training progresses. This observation justifies our decision to focus on analyzing the seconds per iteration from the last round of training (e.g. in Tables 3, 5, 7, 12, and 14).

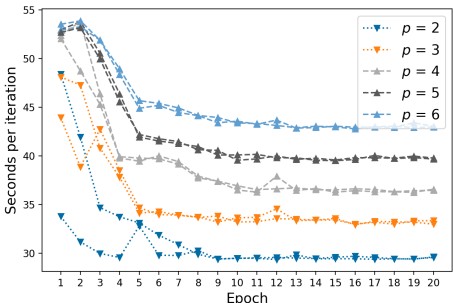

Figure 4: Seconds per iteration associated with clients using various $p^{tr}$ values with FLAME to learn the MRPC task in a FL system using Setting A. We plot these values across each of the 20 total training rounds.

# G   MRPC RESULTS FROM ABLATING THE COLLABORATION OF FLAME

In Table 13, we include results for MRPC associated with the ablation study from Section 5.1. These results support the observation that training a single, centralized model using FLAME leads to compromised AUC score, particularly with late-exiting samples.

# H   SECONDS PER ITERATION PER CLIENT FOR GROUPED BACKPROPAGATION

In Table 7 of Section 6.1, when discussing the grouped backpropagation extension for FLAME, we reported the average seconds per training iteration across clients that used different $p^{tr}$ values. Now, in Table 14, we list the full set of seconds per iteration values that were used to compute the averages.

# I   CLIENT-WISE EXIT LAYER OVER TIME

The design of patience-conscious aggregation is based on the assumption that clients using higher $p^{tr}$ will have training samples exiting later. However, as we mentioned in Section 6.2, we find that this is not always the case. The plots in Figure 5 support this observation. For instance, with SST-2 and MRPC, in early rounds of training, a client with $p^{tr} = 2$ has higher average exit layer than clients with higher $p^{tr}$.

# J   50-CLIENT SENT140 RESULTS COMPARING FLAME TO PRIOR WORKS

These experiments go along with our 25-client experiments in Section 7, but now, we use 50 clients. Again, we are using the Sent140 task and training for 20 rounds. Specifically, we use a set of 50 clients that have the following training dataset sizes: 50, 75, 192, 112, 52, 94, 111, 64, 60, 101, 51, 71, 76, 54, 50, 58, 78, 65, 63, 62, 51, 248, 53, 109, 152, 113, 92, 51, 151, 52, 216, 103, 58, 54, 93, 177, 79, 84, 73, 107, 61, 279, 141, 62, 238, 75, 78, 60, 93, 50. We simply chose a random set of 50 clients that had at least 50 training samples.

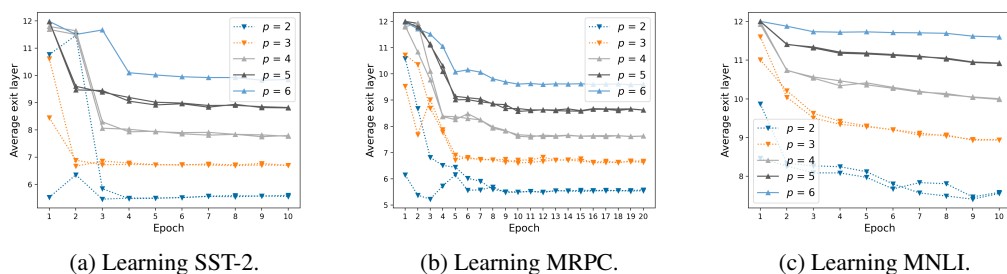

(a) Learning SST-2.     (b) Learning MRPC.     (c) Learning MNLI.

Figure 5: Plotting the average exit layer of local training samples for each of 10 clients across 10 rounds of training in a FL system. Each subplot is associated with learning a different task. Clients are using FLAME with various $p^{tr}$ values (using Setting A).

We compare results for FLAME, HeteroFL, ScaleFL, and AFD, using similar compute budgets. We use $p^{tr} = 3$ for FLAME, which led to an average exit layer of 6.97. Using the formulas in Appendix B, we match this average exit to the following hyperparameters that equate to roughly the same FLOP usage: $r = 0.8$ for HeteroFL, $(d = 9, r = 0.9)$ for ScaleFL, and $\delta = 0.4$ for AFD. The results of these experiments are included in Table 15, which reports separate AUC scores for early- and late-exiting test samples (see Appendix A.5 for details on the early vs. late distinction).

## K FLAME'S MEMORY SAVINGS

Although FLAME's primary intention is to reduce the computational costs associated with training, we find that the method also leads to memory savings. As FLAME allows samples to only pass through subsets of a model's total parameters, activations will only be computed and stored for subsets of the total parameters. Activations typically need to be stored in RAM as samples forward-pass through a network so that they can be used to update parameters during backpropagation, and this can amount to burdensome memory overhead. Therefore, as we see in Table 16, using FLAME can save up to approximately 200 MB of RAM. In this table, we report the total amount of GPU RAM that was used in training 10-client FL systems where all clients use FLAME with the same $p^{tr}$ value. We compare these values to baseline values clients do not use FLAME and therefore do not allow any early exiting. The minimal logical $p^{tr}$ value to use is 2, which is where we see 100 (with MNLI) to 200 (with SST-2 and MRPC) MB in savings. Using the maximum $p^{tr}$ value considered in our paper (6) results in no memory savings, which makes sense since the average exit layer in these experiments is nearly 12 (the same as in the baseline).

---

**Algorithm 1:** FLAME: FEDERATED LEARNING WITH SAMPLE-ADAPTIVE MULTI-EXIT
TRAINING. Each client $m \in \mathcal{M}$ has local data $\mathcal{D}_m$ and a training-time patience value $p_m^{\text{tr}}$. Clients
train for $E$ local epochs across $T$ global rounds with learning rate $\eta$. Models have $L$ hidden
layers, each followed by an internal classifier (IC), which is a linear projection $W_\ell^{\text{IC}} h_\ell$ of the
hidden state into a softmax over classes. We write $\text{Layer}_\ell(h_{\ell-1}; W)$ for the mapping of hidden
state $h_{\ell-1}$ through the $\ell$-th hidden layer (with $h_0 = x$). For now, we assume batch size = 1. See
Algorithm 2 for the grouped backpropagation extension that enables larger minibatch sizes. Note
that we use FedAvg for aggregation by default, but line 8 can be replaced with the aggregation
formulas we introduce in Section 6.2.

1  **Server executes:**
2     Initialize $W_g$;
3     **for** $t = 1$ **to** $T$ **do**
4         **for** *each client* $m \in \mathcal{M}$ **do**
            // send current global parameters
5             $W_m \leftarrow W_g$;
            // client trains locally
6             $W_m \leftarrow$ **ClientUpdate**$(m,\ W_m,\ p_m^{\text{tr}},\ E,\ \eta)$;
7         **end**
        // aggregate local models (using FedAvg by default)
8         $W_g \leftarrow \frac{1}{\sum_{m \in \mathcal{M}} |\mathcal{D}_m|} \sum_{m \in \mathcal{M}} |\mathcal{D}_m|\, W_m$;
9     **end**

10  **Client executes:**
11     **ClientUpdate**$(m,\ W,\ p_m^{\text{tr}},\ E,\ \eta)$:
12         **for** $e = 1$ **to** $E$ **do**
13             **for** *each* $(x, y_{\text{true}}) \in \mathcal{D}_m$ **do**
14                 $c \leftarrow 0$;
                // consecutive-IC-agreement counter
15                 $\hat{y}_{\text{curr}} \leftarrow \bot$;
                // sentinel (no label yet)
16                 $h_0 \leftarrow x$;
17                 **for** $\ell = 1$ **to** $L$ **do**
                    // forward through layer $\ell$
18                     $h_\ell \leftarrow \text{Layer}_\ell(h_{\ell-1}; W)$;
                    // IC (linear projection then softmax)
19                     $z_\ell \leftarrow W_\ell^{\text{IC}} h_\ell$;
20                     $\pi_\ell \leftarrow \text{softmax}(z_\ell)$;
21                     $\hat{y}_\ell \leftarrow \arg\max \pi_\ell$;
22                     **if** $\hat{y}_\ell = \hat{y}_{\text{curr}}$ **then**
23                         $c \leftarrow c + 1$;
24                     **else**
25                         $c \leftarrow 1$;
26                         $\hat{y}_{\text{curr}} \leftarrow \hat{y}_\ell$;
27                   **end**
28                   **if** $c \geq p_m^{\text{tr}}$ *or* $\ell = L$ **then**
29                     $\ell_{\text{exit}} \leftarrow \ell$;
30                     **break**;
31                   **end**
32              **end**
                // compute loss at the chosen exit and update only
                    layers 1:$\ell_{\text{exit}}$ and that IC
33                 $\mathcal{L} \leftarrow \text{CrossEntropyLoss}(z_{\ell_{\text{exit}}}, y_{\text{true}})$;
34                 Update $W_{\leq \ell_{\text{exit}}}$ and $W_{\ell_{\text{exit}}}^{\text{IC}}$ via SGD with learning rate $\eta$ and gradient $\nabla\mathcal{L}$;
35             **end**
36         **end**
37         **return** $W$

---

**Algorithm 2:** GROUPED BACKPROPAGATION. A modification of the client-side local training that happens with FLAME, which allows the use of minibatches with a size $b > 1$. This function, **GroupedClientUpdate**, replaces **ClientUpdate** in the standard FLAME pipeline in Algorithm 1 (where $b = 1$). Note that lines lines 10–26 in **GroupedClientUpdate** are essentially identical to lines lines 16–30 in **ClientUpdate**.

1  **GroupedClientUpdate**$(m, W, p_m^{\text{tr}}, E, \eta, b)$:
2    **for** $e = 1$ **to** $E$ **do**
3      **for** *each minibatch $\mathcal{B} \subset \mathcal{D}_m$ of size $b$* **do**
4        **for** $\ell = 1$ **to** $L$ **do**
5          Initialize exit-layer loss list $\mathcal{L}_\ell \leftarrow []$
6        **end**
       // individually forward-pass samples with patience-based early exits
7        **for** *each $(x, y_{\text{true}}) \in \mathcal{B}$* **do**
8          $c \leftarrow 0$;
         // consecutive-IC-agreement counter
9          $\hat{y}_{\text{curr}} \leftarrow \bot$;
         // sentinel (no label yet)
10          $h_0 \leftarrow x$;
11          **for** $\ell = 1$ **to** $L$ **do**
          // forward through layer $\ell$
12           $h_\ell \leftarrow \text{Layer}_\ell(h_{\ell-1}; W)$;
          // IC (linear projection then softmax)
13           $z_\ell \leftarrow W_\ell^{\text{IC}} h_\ell$;
14           $\pi_\ell \leftarrow \text{softmax}(z_\ell)$;
15           $\hat{y}_\ell \leftarrow \arg\max \pi_\ell$;
16           **if** $\hat{y}_\ell = \hat{y}_{\text{curr}}$ **then**
17             $c \leftarrow c + 1$;
18           **else**
19             $c \leftarrow 1$;
20             $\hat{y}_{\text{curr}} \leftarrow \hat{y}_\ell$;
21          **end**
22           **if** $c \geq p_m^{\text{tr}}$ *or* $\ell = L$ **then**
23             $\ell_{\text{exit}} \leftarrow \ell$;
24             $\mathcal{L}_{\text{curr}} \leftarrow \text{CrossEntropyLoss}(z_{\ell_{\text{exit}}}, y_{\text{true}})$;
25             store $\mathcal{L}_{\text{curr}}$ in $\mathcal{L}_{\ell_{\text{exit}}}$;
26             **break**;
27           **end**
28         **end**
29        **end**
       // perform one backpropagation per non-empty exit-layer group
30        **for** $\ell = 1$ **to** $L$ **do**
31          **if** $\mathcal{L}_\ell \neq []$ **then**
32            $\mathcal{L}_\ell^{\text{avg}} \leftarrow \frac{1}{|\mathcal{L}_\ell|} \sum_{\lambda \in \mathcal{L}_\ell} \lambda$;
33            Update $W_{\leq \ell}$ and $W_\ell^{\text{IC}}$ via SGD with learning rate $\eta$ and gradient $\nabla \mathcal{L}_\ell^{\text{avg}}$;
34          **end**
35        **end**
36      **end**
37    **end**
38    **return** $W$;

---

Table 12: Seconds per iteration from the last iteration of training for clients using various $p^{tr}$ values with FLAME in Setting A. We compare these values against a baseline where FLAME is not used (no early exiting occurs during training).

| Task | # samples per client | $p^{tr}$ | Avg. secs/it | % change from baseline |
|------|------|------|------|------|
| MRPC | 1000 | Baseline | 55.31 | – |
|  |  | 2 | 29.63 | -46.43% |
|  |  | 3 | 33.37 | -39.67% |
|  |  | 4 | 36.59 | -33.85% |
|  |  | 5 | 39.79 | -28.06% |
|  |  | 6 | 43.04 | -22.18% |
| MNLI | 15000 | Baseline | 854.75 | – |
|  |  | 2 | 614.93 | -28.06% |
|  |  | 3 | 673.37 | -21.22% |
|  |  | 4 | 776.67 | -9.13% |
|  |  | 5 | 814.41 | -4.72% |
|  |  | 6 | 840.26 | -1.70% |

Table 13: Evaluation AUC scores resulting from training a single, centralized model on the MRPC task using MET with various $p^{tr}$ values. We report scores separately for early- and late-exiting evaluation samples, defined by the exit layer taken during inference with $p^{ev} = 4$ (thresholds are detailed in Appendix A.5). We also include results from a baseline model that did not use MET (no early exiting allowed).

| Method | Samples | AUC |
|------|------|------|
| Baseline | Early-exiting | 0.82 |
|  | Late-exiting | 0.71 |
| MET, $p^{tr} = 2$ | Early-exiting | 0.83 |
|  | Late-exiting | 0.58 |
| MET, $p^{tr} = 3$ | Early-exiting | 0.83 |
|  | Late-exiting | 0.65 |
| MET, $p^{tr} = 4$ | Early-exiting | 0.84 |
|  | Late-exiting | 0.67 |

Table 14: Seconds per training iteration for clients using different $p^{tr}$ values with FLAME to learn the SST-2 task in a FL system that uses Setting A. The clients are using backpropagation with FLAME and we report results associated with different grouping strategies and $b$ values. We compare these results to those from a baseline where FLAME is used without grouped backpropagation.

| Grouping strategy | $b$ | Seconds/iteration | | | | |
|---|---|---|---|---|---|---|
| | | $p^{tr} = 2$ | $p^{tr} = 3$ | $p^{tr} = 4$ | $p^{tr} = 5$ | $p^{tr} = 6$ |
| Standard FLAME | 1 | 102.90 | 120.09 | 134.41 | 147.43 | 160.06 |
| Proposed grouping | 16 | 70.09 | 82.99 | 93.72 | 103.68 | 113.73 |
| Proposed grouping | 32 | 90.23 | 107.73 | 120.26 | 127.80 | 132.73 |
| Proposed grouping | 64 | 62.30 | 78.61 | 112.01 | 120.42 | 125.22 |
| Proposed grouping | 128 | 65.58 | 78.11 | 125.95 | 131.70 | 136.09 |
| Full group | 8 | 72.69 | 86.35 | 99.54 | 110.45 | 118.79 |
| Full group | 32 | 81.24 | 92.48 | 102.88 | 112.95 | 120.12 |
| Binary grouping | 32 | 61.65 | 78.32 | 89.55 | 99.63 | 108.23 |
| Random grouping | 32 | 67.39 | 80.08 | 92.57 | 102.73 | 110.92 |
| Distant pairing | 32 | 83.91 | 87.03 | 101.89 | 111.61 | 118.43 |
| Close pairing | 32 | 78.07 | 86.51 | 101.93 | 111.66 | 118.78 |

Table 15: Comparing FLAME, HeteroFL, ScaleFL, and AFD final global model evaluation AUC scores on the Sent140 task. AUC scores are shown separately for early- and late-exiting samples, defined by the exit layer taken during inference with $p^{ev} = 4$ (thresholds provided in Appendix A.5). We train for 20 rounds with 50 clients that all use the same values for $p^{tr}/r/\delta$. Methods are compute-matched to similar FLOPs (see Appendix B). FLAME enables inference-time early exiting with $p^{ev} = 4$. The other methods are not designed to train ICs so they do not enable any early exiting during evaluation.

| Method | Parameter | AUC | | Average GFLOPs/client |
|---|---|---|---|---|
| | | Early samples | Late samples | |
| FLAME | $p^{tr} = 3$ | **0.8110** | **0.7161** | 84.23 |
| HeteroFL | $r = .8$ | 0.7034 | 0.6144 | 92.77 |
| ScaleFL | $d = 9, r = .9$ | 0.7268 | 0.5153 | 88.06 |
| AFD | $\delta = .4$ | 0.5346 | 0.4800 | 86.97 |

Table 16: Average exit layer and total GPU RAM used in training FL systems where are clients are using FLAME with the same $p^{tr}$ value. We compare these values to baseline where none of the clients use FLAME (no early exiting allowed during training). Note that we used an NVIDIA A100-SXM4-40 GB GPU and we only had access to RAM usage metrics that are rounded to the nearest 0.1 GB.

| Task | # samples per client | $p^{tr}$ | Avg. exit layer | GPU RAM (GB) |
|------|------|------|------|------|
| SST-2 | 4000 | Baseline | 12 | 16.0 |
|  |  | 2 | 7.712 | 15.8 |
|  |  | 6 | 11.914 | 16.0 |
| MRPC | 1000 | Baseline | 12 | 16.0 |
|  |  | 2 | 9.987 | 15.8 |
|  |  | 6 | 11.998 | 16.0 |
| MNLI | 1500 | Baseline | 12 | 16.0 |
|  |  | 2 | 9.564 | 15.9 |
|  |  | 6 | 11.999 | 16.0 |

