# OpenReview forum: "FLAME: Reducing Computation in Federated Learning via Sample-Adaptive Multi-Exit Training"
_ICLR.cc/2026/Conference — Submitted to ICLR 2026_

### Official Review · Reviewer_fRv4 · 2025-10-23

**Soundness:** 2
**Presentation:** 1
**Contribution:** 2
**Rating:** 2
**Confidence:** 3

**Summary:**

The paper introduces FLAME, a novel sample-adaptive multi-exit training framework for Federated Learning (FL). Unlike traditional methods that allocate fixed or client-specific subnetworks, FLAME dynamically adjusts computation per sample through early-exit classifiers. Each input can terminate forward propagation early if successive internal classifiers agree, thereby saving compute while maintaining accuracy.

**Strengths:**

1. The work is well motivated, addressing a genuine bottleneck in FL: computation heterogeneity rather than storage limitations. By focusing on resource-constrained clients such as those in IoT environments, the paper demonstrates clear practical relevance and positions its contribution as directly applicable to real-world federated systems.

2. The work is supported by comprehensive experiments conducted on multiple NLP benchmarks, including SST-2, MRPC, MNLI, and Sent140. These experiments are complemented by detailed ablation and scalability studies that validate the approach and highlight its effectiveness in reducing computation without degrading performance.

3. The inclusion of a convergence proof adapted from FedPMT provides solid theoretical grounding, ensuring that the proposed method is not only empirically effective but also theoretically sound. This strengthens the work’s rigor and reassures readers about its stability in federated optimization.

**Weaknesses:**

1. The idea of incorporating Multi-Exit mechanisms into federated learning is not novel. Several prior works have already explored this direction [1–3]. However, the authors neither compare FLAME with these existing approaches nor introduce any substantive refinement to the Multi-Exit paradigm to better adapt it to the federated learning setting.
[1] Lee, Royson, et al. "Recurrent Early Exits for Federated Learning with Heterogeneous Clients." Forty-first International Conference on Machine Learning.
[2] Qu, Lehao, et al. "DarkDistill: Difficulty-Aligned Federated Early-Exit Network Training on Heterogeneous Devices." Proceedings of the 31st ACM SIGKDD Conference on Knowledge Discovery and Data Mining V. 2. 2025.
[3] Ilhan, Fatih, Gong Su, and Ling Liu. "Scalefl: Resource-adaptive federated learning with heterogeneous clients." Proceedings of the IEEE/CVF Conference on Computer Vision and Pattern Recognition. 2023.

2. The overall presentation of the paper requires improvement. The organization is confusing. For example, the authors introduce experimental settings before presenting the design of their method, which disrupts the logical flow of the paper. In addition, the design section includes several experimental results that would be more appropriately placed in the evaluation section. Furthermore, there are visible blank spaces in Table 2, suggesting incomplete or improperly formatted content. These issues collectively reduce the paper’s readability and professionalism.

3. The experimental scope of the paper is quite limited, as all evaluations are confined to BERT-based NLP tasks. It remains unclear whether the sample-adaptive multi-exit strategy would function effectively in architectures such as CNNs or Vision Transformers (ViTs), where layer semantics and computational dynamics differ significantly from those in Transformer-based NLP models. Extending the evaluation to diverse modalities would greatly strengthen the paper’s claims about the broad applicability and robustness of FLAME.

4. The literature review in this paper is insufficient, as most of the cited works are relatively outdated. In addition, the chosen baseline methods are also based on older approaches, which limits the paper’s ability to demonstrate progress over the current state of the art. Incorporating more recent studies and stronger baselines would significantly improve the completeness and credibility of the work.

5. The authors propose the use of a multi-exit mechanism based on device resource limitations; however, they only consider computational constraints while neglecting storage constraints. In practice, each device is still required to deploy the entire large-scale machine learning model, which undermines the claimed advantage for resource-limited clients. A truly resource-efficient design should also account for memory and storage requirements, not just computation.

**Questions:**

1. The claimed contribution lacks clear differentiation from prior research. The integration of multi-exit mechanisms into federated learning has already been explored in several earlier studies. The authors should clearly explain how their approach advances beyond these previous efforts and specify what unique advantages or improvements FLAME provides compared with existing multi-exit-based FL methods.

2. The comparison baselines used in the paper are relatively outdated. The authors primarily evaluate against earlier methods rather than the latest advances in federated learning efficiency or adaptive computation. To provide a fair and convincing assessment of FLAME’s effectiveness, the paper should include comparisons with more recent and competitive baselines that reflect the current state of the art.

---

### Official Review · Reviewer_agcu · 2025-10-25

**Soundness:** 2
**Presentation:** 1
**Contribution:** 2
**Rating:** 2
**Confidence:** 4

**Summary:**

This paper tackles the limitation in existing solutions to system heterogeneity in federated learning, where clients train local models of a fixed size regardless of the input sample. To overcome this issue, it introduces a mechanism that dynamically determines when to perform an early exit based on each sample. Specifically, the method allows a sample to exit early when consecutive internal classifiers agree on the prediction; otherwise, the model continues training deeper layers. Through this process, the framework adaptively adjusts the effective size of each sub-model, leading to more efficient computation.  Through a series of ablation studies, the paper attempts to demonstrate the effectiveness of this sample-adaptive multi-exiting mechanism.

**Strengths:**

The idea of adjusting early exits based on each sample, rather than applying the same sub-model to all data, is plausible.

The paper provides several ablation studies to demonstrate the effectiveness of the proposed sample-adaptive multi-exiting approach.

The paper provides a detailed analysis of the proposed method in terms of implementation details such as batch size and the aggregation rule.

**Weaknesses:**

**The explanation of the experimental setup is not sufficiently detailed, making it difficult to interpret and trust the reported results.** For example, in Table 2, the baseline does not employ early exit, which suggests that it trains the full model. If so, it should represent the upper bound of performance. It is therefore unclear why its performance is lower than that of FLAME. More specific questions are listed in the Questions below.

**The achievable computation cost savings are limited, and there are no communication cost savings.** Unlike the baselines, the proposed method still communicates the entire full model, resulting in no reduction in communication overhead. As shown in Table 3, even the computation cost savings are relatively modest, up to around 50% at best.

**The necessity of deeper layers is questionable.** In Table 4, the baseline, despite having no early exit, shows a lower late-exiting AUC. Moreover, when early exit is allowed, deeper layers are trained with fewer samples, which further amplifies this discrepancy. Wouldn’t it be more stable and efficient to have all clients train smaller models without the deeper layers altogether?

**Questions:**

In Table 2, why is the avg. exit value for the baselines without early exiting not 12? If early exit is not applied, inference should always be performed using the full model. Isn’t the avg. exit therefore expected to be equal to the maximum depth (i.e., 12)?

In Table 5, it’s unclear why the performance of the fixed exit layer setting is so low. In particular, if all clients exit at layer 11, that effectively means they have trained the full model. How can this configuration perform worse than FLAME?

In Figure 2, the difference between the higher-patience client (b) and the lower-patience client (a) does not appear to increase noticeably in the later layers. What could be the reason for this?

In Table 8, HeteroFL does not incorporate early exit. How, then, was its evaluation conducted in this setting?

---

> ### Author Response · Authors · 2025-11-21
> **Response to reviewer agcu**
>
> Thank you for taking the time to read our work and provide this helpful feedback. We are providing separate comments to answer your questions and address the weaknesses you’ve identified. We have also uploaded a new version of the paper that incorporates your feedback (we explain the paper edits in our comments below).

---

> ### Author Response · Authors · 2025-11-21
> **Answers to the questions asked**
>
> **Q1 (clarifying inference-time early exiting use with baselines):**
>
> For Table 2, we are using inference-time p^{ev} = 4 for all models. Therefore, even though the baseline doesn’t allow early exiting during training, it allows early exiting during inference. This is actually the way that this multi-exit mechanism was first introduced in [1,2] (concurrent works), where training would happen as normal, but early-exiting would be enabled during inference. We did mention in the caption of Table 2 that p^{ev} = 4, but we have edited the caption of Table 2 to make this even more clear that this applies to all settings, including the baseline. We also explained in Section 3 in the “Evaluation metrics and protocols” paragraph that evaluation would use p^{ev} = 4 (unless otherwise noted), but we understand that this may be easy to miss, so we’ve added to a sentence in paragraph 3 of Section 4 (“For comparison, we include baselines…do allow inference-time early exiting”) to make it extra clear that early exits are being used during inference for Table 2.
>
> [1] Yigitcan Kaya, Sanghyun Hong, and Tudor Dumitras. Shallow-Deep Networks: Understanding and mitigating network overthinking. In Proceedings of the 2019 International Conference on Machine Learning (ICML), Long Beach, CA, Jun 2019.
> [2] Wangchunshu Zhou, Canwen Xu, Tao Ge, Julian McAuley, Ke Xu, and Furu Wei. Bert loses patience: Fast and robust inference with early exit, 2020
>
> **Q2 (explaining the inferiority of using fixed exit layers):**
>
> This logic does make sense when you compare fixed exit layer 11 (AUC = .846) and FLAME with p^{tr} = 2 (AUC = .774, average exit layer = 6.403). Here, it is likely that FLAME’s lower AUC score is due to the fact that only about half of the network is receiving updates. However, with FLAME and p^{tr} = 3, now the average exit is 9.984 and most of the model is being updated during training, which results in a higher AUC (0.877). The reason why this FLAME AUC is higher than the AUC that used fixed exit layer 11 relates to the fact that 9.984 isn’t a fixed exit layer, it’s an average. Therefore, some samples are actually passing through the full network and updating all network parameters, but some are exiting earlier in the network. FLAME is a strategy for choosing when it is best to force a sample to pass through the full network vs. when we can save on computation and exit early. In short, FLAME with p^{tr} = 3 has a similar average exit to using fixed exit layer 11, but FLAME strategically has some samples exit later while other samples save costs and exit earlier, in a way that can benefit AUC compared to forcing all samples to exit at the same layer. In the new version of the paper, we have revised the end of Section 5.2 to explain this scenario and reasoning more clearly (starting with “This shows that some samples, presumably the more difficult ones…”).
>
> **Q3 (fixing the inclusion of an incorrect plot):**
>
> Thank you for pointing this out. This helped us realize that we accidentally included the wrong version of our plots (2b was correct, but 2a was not). We have now fixed Figure 2 to show the correct plots, which should make the pattern more obvious. Now, you can see that with the p^{tr}=2 client, layers 10,11, and 12 have substantially less variance than they do with the p^{tr}=6 client, especially as training progresses. This makes sense because the lower-patience client, while allowing many samples to exit before reaching layers 10-12, will not be updating those layers’ parameters nearly as often as the higher-patience client.
>
> **Q4 (clarifying the use of early exits when comparing to other solutions):**
>
> Thank you for pointing out this lack of clarity. We have added a sentence to the paper (please see paragraph 2 of Section 7, “FLAME uses p^{ev} = 4 during evaluation, but…”) and have modified the phrasing in the Table 8 caption (“FLAME enables inference-time early exiting with p^{ev}=4, but…”) to make this more clear. For FLAME, we were evaluating the global model using p^{ev} = 4, but with the other methods (HeteroFL, ScaleFL, and AFD), we were just evaluating the full global model without any early exiting. The non-FLAME methods do not train any ICs so they cannot enable early exiting.

---

> ### Author Response · Authors · 2025-11-21
> **Addressing the identified weaknesses**
>
> **W1 (explaining FLAME’s counterintuitive AUC advantage over the baseline):**
>
> Even though it may intuitively seem like the baseline would have the best AUC, we find that this isn’t always the case. We think this is very likely due to one or both of the following issues.
>
> 1. Overfitting. For instance, with Table 2, we see that the baseline performs worse than FLAME with SST-2, which happens to be the easiest task to learn out of the three. Therefore, when we train the full model and don’t allow early exiting, the model is likely overfitting on the training data and hence shows lower evaluation AUC because it is not generalizing well to new data. Overfitting is known to occur when a model has excessive capacity for learning relatively simple tasks.
>
> 2. Overthinking. We also see in Table 2 that the SST-2 baseline model has later average exit during evaluation when using p = 4. [1] introduces the overthinking problem, which is the phenomenon where some samples receive incorrect predictions if the model is too deep and does excessive computation for relatively easy-to-predict samples. This is why [1] introduces the idea of early exiting during inference. It shows that allowing samples to exit early can lead to more accurate predictions. So, since FLAME-trained models lead to earlier exiting during inference, they are leading to more accurate predictions than the baseline, which faces the overthinking problem.
>
> We have added a short explanation of this in paragraph 3 of Section 4 (starting with “We suspect that this could be due to two reasons…”).
>
> [1] Yigitcan Kaya, Sanghyun Hong, and Tudor Dumitras. Shallow-Deep Networks: Understanding and mitigating network overthinking. In Proceedings of the 2019 International Conference on Machine Learning (ICML), Long Beach, CA, Jun 2019.
>
> **W2 (justifying the lack of communication cost savings):**
>
> We do acknowledge the fact that it may seem like FLAME has the drawback of requiring the communication of all model parameters, but our work focuses on reducing computation cost instead of communication costs. Either FLAME will be most useful when computational costs are a concern while communication cost is not, or FLAME could be paired with some separate method that focuses on reducing communication costs (which could be an interesting direction for future study). We have added an additional sentence to the conclusion (starting with “Although this paper focuses on saving computational costs,…”) that mentions this opportunity for future research. We have also taken out a sentence from the introduction (“Appendix J also reports minor memory savings.”) so that we do not mislead readers into thinking that FLAME intends to make significant memory savings. We want it to be clear that this work is focused on saving computational costs, but we do think it is still interesting to report the very minor inadvertent memory savings.
>
> Regarding your concern about the computational savings not being substantial enough, we think that achieving ~50% computation savings while maintaining competitive accuracy is actually a significant feat. For instance, reducing compute by half can double the battery lifetime for some small client device.
>
> **W3 (the advantage of using sample-adaptiveness over smaller architectures):**
>
> If we understand your concern correctly, then having all clients train smaller models would essentially be equivalent to having all samples exit early. For instance, having a smaller, 7-layer model instead of a 12-layer model would be equivalent to forcing all samples to exit at layer 7 in a 12-layer model. And we have these results in Table 5, where the bottom half of the table lists AUC scores associated with models where every sample was forced to exit at some early layer during training. Here, we see that the fewer layers that the samples have to pass through, the worse the resulting AUC score is during evaluation. This suggests that training smaller models will sacrifice AUC.
>
> The advantage of the early exit mechanism is that, when we consider its use during inference, it allows each individual sample to only pass through the network up until the model is confident enough in its prediction for the sample. Many of our paper’s results show that the average exit taken during inference is close to the 9-12 range. This means that many samples are needing most/all of the 12 model layers in order to receive a confident prediction, and would likely not be accurately predicted if we restricted all clients to a smaller model.
>
> Even if, hypothetically, no samples would need more than 7 layers to be accurately predicted and we therefore only train models with 7 hidden layers, then there still would be some samples that can benefit from using even fewer layers, which is why FLAME is useful. Samples can utilize however many layers they need, but we need to make sure that even the difficult samples have sufficient layers to potentially pass through.

---

> ### Comment · Reviewer_agcu · 2025-11-24
>
> Thank you for the detailed answers to my previous questions. However, some aspects of the experimental settings and results still remain unclear.
>
> In Table 5, the performance gap between exiting at layer 10 and layer 11 appears to be quite large. Could you elaborate on how this difference should be interpreted?
> In addition, regarding your response to W3, I would like to clarify my understanding: for example, if the model was trained with a fixed exit at layer 7, shouldn’t it also perform inference with a fixed exit at the same layer?
> From your explanation, it seems that the inference stage determines the exit layer using a common criterion, without considering the specific configuration used during training.
> If possible, I would like to see results where inference is performed only using the internal classifier that was actually used during training, so that the consistency between training and inference can be more clearly evaluated.
>
> There appears to be a case where the manuscript cites a non-existent or fabricated paper. This issue should be carefully addressed, as it raises concerns about the validity of the reference list.

---

> > ### Author Response · Authors · 2025-11-24
> >
> > Thank you for this followup. We appreciate this opportunity to discuss these interesting questions that you are raising.
> >
> > The large difference between exiting at layers 10 and 11 is probably due to the fact that a large number of samples need later layers for some sort of processing that is crucial to the model’s decision. For instance, maybe there is some crucial processing that happens at layer 11. About 50% of samples may be able to be accurately predicted before then (hence the ~.50 AUC with exiting at layers 6-10), but there may be ~30% of samples that need that 11th layer in order to be accurately predicted (hence that jump in AUC score when the exit is pushed back to layer 11). Figure 3 supports this idea. For instance, with Figure 3b, we see most samples exiting at either layer 6, 10, or 11, and basically no samples exit at any other layer. This means that the model can confidently/accurately predict some samples after only processing it through the earlier layers, but others need the last couple of layers’ processing. The key observation here is that there is a large cluster of samples that can exit early, which is why exiting at 6-10 results in roughly the same AUC, but there is another large cluster of samples that need to exit late, which is why adding the 11th layer of processing results in a jump in AUC.
> >
> > We will run these additional experiments that you are describing regarding W3 and add a new followup comment with those results as soon as we have them. If we understand you correctly, this would involve fixing the exit at some layer during training, e.g. layer 7, and then during inference, only allowing samples to exit at layer 7.
> >
> > We also just found and corrected the reference that included the wrong information (the DepthFL reference had the wrong authors/venue). Thank you for catching this error. We have double-checked the other references as well and believe that there are no more errors.

---

> > > ### Author Response · Authors · 2025-12-02
> > > **Additional results regarding W3**
> > >
> > > We have now finished a new set of experiments as suggested. We also added these results to a new version of the paper that is now uploaded. These new results are in Table 6, which is introduced and explained in Section 5.2.
> > >
> > > Here, for models that force all samples to exit at the same pre-determined layer during training, we don't allow any exiting to happen beyond this layer during inference. The revised version of the paper explains this further. We now use MRPC instead of SST-2. This new set of results still supports our claim that it is advantageous to allow samples to exit at different layers instead of forcing them to all exit at the same layer. We can see that FLAME leads to higher AUC compared to the fixed-exit models when using similar compute. For instance, when FLAME uses p^{tr} = 4 during training, the resulting AUC is 0.738 the average exit ends up being 10.854 during training, which can be rounded to 11. However, when we force *all* samples to exit at layer 11 during training, the resulting AUC score is lower: 0.711.
> > >
> > > Note that, for the fixed-exit models, although we do not allow inference-time exiting after the layer that samples exited through during training, we do still allow exiting to happen before that layer. We are still using patience-based exiting with p^{ev}=4. All of the ICs before the training-time exit layer are still being optimized so we believe it is still logical to allow their use for early exiting during evaluation.
> > >
> > > | Exit strategy         | AUC   | Avg. exit (train) | Avg. MFLOPs/sample/iter |
> > > |-----------------------|-------|--------------------|---------------------------|
> > > | FLAME, \(p^{tr}=2\)   | 0.765 | 6.774              | 81859                    |
> > > | FLAME, \(p^{tr}=3\)   | 0.713 | 10.700             | 129302                  |
> > > | FLAME, \(p^{tr}=4\)   | 0.738 | 10.854             | 131163                  |
> > > | FLAME, \(p^{tr}=5\)   | 0.755 | 11.752             | 142015                  |
> > > | FLAME, \(p^{tr}=6\)   | 0.760 | 11.906             | 143876                  |
> > > | Exit layer 6          | 0.691 | 6                  | 72506                    |
> > > | Exit layer 7          | 0.708 | 7                  | 84590                    |
> > > | Exit layer 8          | 0.712 | 8                  | 96674                    |
> > > | Exit layer 9          | 0.727 | 9                  | 108759                   |
> > > | Exit layer 10         | 0.721 | 10                 | 120843                   |
> > > | Exit layer 11         | 0.711 | 11                 | 132927                   |

---

### Official Review · Reviewer_ejMK · 2025-11-01

**Soundness:** 3
**Presentation:** 3
**Contribution:** 3
**Rating:** 6
**Confidence:** 4

**Summary:**

This paper introduces FLAME (Federated Learning with sample-Adaptive Multi-Exiting), which reduces the training cost at clients of Federated Learning by using sample-adaptive early stopping in local training of clients. The method allows each sample to exit the neural network at the earliest layer where the model can confidently predict its output, thus saving computational resources without sacrificing accuracy. That is, after one or more hidden layers of the model, the sample is passed through an internal classifier (IC). The forward pass will stop at the point where p successive ICs have the same classification. Although the method does not reduce the memory footprint at each client, it could reduce the computation cost for easy samples since they are performed in a smaller subnetwork.

**Strengths:**

S1. FLAME help to improve the performance, e.g., AUC, and reduce the training cost, yet reduce the time per iterations.
S2. The introduction of sample-adaptive early exits in FL is a novel approach.

**Weaknesses:**

W1. FLAME requires clients to store the entire model, even though they may not utilize all of its layers. Consequently, FLAME may still be infeasible for training large models on resource-constrained clients, as the overall memory footprint is not reduced. Specifically, although the authors mentioned the memory reduction in Table 14 (Appendix), the reduction is trivial.

W2. The findings presented in the paper are primarily based on empirical results, without any accompanying theoretical analysis. Moreover, the empirical evaluation focuses on a limited range of settings (in terms of models and datasets) and employs a non-typical federated learning setup—with only 10 clients and unclear details on how the local datasets are constructed. As a result, the proposed idea appears to lack generality.

**Questions:**

Q1. Please provide a detailed discussion on how the model is updated in FLAME with grouped backpropagation. Explain why the grouping strategy leads to a reduction in AUC. Additionally, in Table 6, the authors should compare FLAME with the setting (No FLAME + b > 1). In this case, does FLAME help reduce the total local training time? Furthermore, note that performing the forward pass sample by sample (instead of by batch) may increase the training time—this point should also be discussed.

Q2. The performance of FLAME largely depends on how the parameter p is chosen. However, in the 10 settings presented in the paper (Table 1), p is fixed for each client. Moreover, using the same fixed value of p across all clients may not be optimal. Please discuss how p should be selected in practice.

Q3. The results in Table 2 show that FLAME under-performs in certain cases. Please discuss which datasets or tasks FLAME performs better or worse on compared to the baseline methods.

---

> ### Author Response · Authors · 2025-11-21
> **Response to reviewer ejMK**
>
> We appreciate your time and constructive feedback. We respond to each of your questions and concerns in the comments below. We have also uploaded a new version of the paper that incorporates your feedback.

---

> ### Author Response · Authors · 2025-11-21
> **Answers to the questions asked**
>
> **Q1 (explaining grouped backpropagation):**
>
> With grouped backpropagation, during training, samples still forward pass through the network individually. However, they do not backpropagate through the network individually. Once a full batch of samples forward-passes through the network, they will be grouped based on their chosen exit layer. For instance, with our proposed grouping strategy, we will just have one group per possible exit (for a 12-layer model, this equates to a maximum of 12 possible groups). Then, for each nonempty group, all of the losses associated with the samples in that group will be averaged and then the average loss is what’s backpropagated through the network, starting from the group’s exit layer. For example, if we are using batch size 32, then each sample in the batch will forward-pass through the network individually, but the backpropagation will only happen once for every group. So, the backpropagation happens at most 12 times for the full batch, but often significantly less than 12 times since we find many layers don’t have any samples exiting at them (see Figure 3 for evidence of this). Please see Algorithm 2 of Appendix C if you would like to see our pseudocode for the grouped backpropagation process and please let us know if there is anything more we can clarify about this process. We have modified a sentence in Section 6.1 of the paper, switching from saying "aggregated" to “averaged”, in order to hopefully make the process more clear.
>
> One possible reason for the AUC reduction that grouped backpropagation causes is the fact that learning the SST-2 task with a BERT model, when using normal minibatch gradient descent, tends to lead to higher performance when smaller batch sizes are used. We choose to use relatively large batch sizes for grouped backpropagation because the batches get split into smaller groups (i.e. one per exit layer). Using a batch size that’s too small may mean that multiple exit layers only have one sample assigned to their backpropagation groups, meaning that the backpropagation happens separately for an individual sample and we thereby don’t reap the savings that grouped backpropagation is designed to allow. However, when we use a larger batch size in order to avoid groups with only one sample, we are also likely causing some groups to get decently large (as supported by Figure 3 showing that many samples tend to cluster at specific exit layers). Perhaps the reduced AUC score is due to the backpropagation sometimes happening in groups that are larger than ideal.
>
> **Q2 (guidance for patience selection):**
>
> One nice thing about FLAME, especially compared to many prior solutions, is that it allows clients to very easily adjust how much compute they want to use during training. We do not have any concrete guidelines for choosing p values. However, one safe strategy for clients unsure of which p value to use could be to start with a low p value (e.g. 3, or even 2 if they want to be as modest as possible), and then once they train for a round or two and realize that they have more compute they are able to allot to training, they can very easily start using a higher p value. There are no extra steps required for changing the patience they use. It is just a simple value re-assignment. If a client ever needs to start using less compute for some reason, they can easily decrease their patience value. Basically, a client should just keep an eye on how much compute they’re using compared to how much they have available to use, and can easily adjust their p values accordingly. We have added an extra sentence at the end of the second paragraph of Section 4 in order to offer a little more guidance regarding patience selection (starting “Therefore, even though we do not have any concrete instructions or heuristics…”).
>
> **Q3 (explaining cases where FLAME underperforms):**
>
> Out of the three tasks in Table 2, FLAME only underperforms compared to the baseline with MRPC and we think it is strongly likely that this is due to the fact that MRPC is a relatively difficult-to-learn task (e.g., on the GLUE benchmark leaderboard, the MRPC performances are consistently significantly worse than those of SST-2) and it has such a small training dataset. It has 3069 samples total, which, when split between 10 clients amounts to only 306 samples per client. By comparison, SST-2 has 67349 training samples total and MNLI has 392702.

---

> ### Author Response · Authors · 2025-11-21
> **Addressing the identified weaknesses**
>
> **W1 (lack of memory savings):**
>
> We understand that FLAME’s requiring of clients to store the full model can be a limitation, especially since the (inadvertent) memory savings that FLAME provides are relatively minor. However, the primary focus of our work is on reducing computational costs for clients and not on reducing storage requirements. We believe this to be reasonable for a couple reasons.
>
> First, recent work has suggested that the growth in computational cost requirements will be more concerning than the growth of memory requirements, hence our interest in specifically tackling computational expense. For instance, [1] shows that their Chinchilla model outperforms the SoTA Gopher model that is 4x larger than Chinchilla but has 4x less training. Their work ultimately suggests that, when you have some fixed compute budget, it’s better to train a smaller model on more data than to strive for a larger model size and less data. This therefore supports the idea that, for ML models, compute is becoming a bigger issue than memory. Although we had briefly talked about this at the end of the first paragraph of the introduction, we have now modified this mention in our revised paper so that the justification for focusing on compute is better explained and supported (the added portion starts with “For instance, Hoffmann et al. (2022) show…”).
>
> Second, FLAME is a method that can easily be paired with other algorithms. For instance, an interesting direction for future study could be to set up a FL system that uses some existing method for deriving smaller sub-networks that clients can train with (e.g. HeteroFL), but that also has clients using FLAME while training those sub-networks, thereby maximizing savings for both memory and compute. In the revised version of our paper, in the conclusion, we have added a brief discussion of the storage requirement limitation (starting with “Although this paper focuses on saving computational costs,…”).
>
> We have also taken out a sentence from the introduction (“Appendix J also reports minor memory savings.”) so that we do not mislead readers into thinking that FLAME intends to make significant memory savings. We want it to be clear that this work is focused on saving computational costs, but we do think it is still interesting to report the very minor inadvertent memory savings.
>
>
> [1] Jordan Hoffmann et al. Training compute-optimal large language models, 2022.
>
> **W2 (reliance on limited empirical results):**
>
> We would like to clarify that our work does include theoretical analysis. We prove the convergence of FLAME, specifically, with a convergence rate that is asymptotically equivalent to the convergence rate of FedAvg. This is stated in the first paragraph of Section 4, but Appendix D has the full proof. We believe that this convergence analysis is an appropriate theoretical contribution for our work. In fact, many related works (e.g. HeteroFL, ScaleFL, InclusiveFL, etc.) limit their theoretical contribution to a convergence guarantee or else exclude any theoretical component.
>
> Regarding the limited number of clients, we would like to point out the fact that we do use a larger pool, 25 clients, in the experiments in Section 7. Since numerous FL papers utilize no more than 10-20 clients in their experiments (e.g. [2-6]), we believe that 10 and 25 are fair amounts of clients to use. Especially since our main goal is to demonstrate the effectiveness of using FLAME compared to other approaches and we think that the theoretical convergence analysis that we include can support the idea that FLAME’s efficacy can generalize to larger (or smaller) FL systems. However, we do understand and agree that it could be valuable to see an even larger-scale setup, so as reviewer veS5 suggested, we are attempting to run some 100-client experiments to add to Section 7. If we are able to successfully finish these experiments before the discussion period ends, then we will add the result to the paper and notify you of the change through an additional comment.
>
> [2] Tao Lin et al. Ensemble distillation for robust model fusion in federated learning, 2021.
>
> [3] Zexi Li et al. Revisiting weighted aggregation in federated learning with neural networks, 2023.
>
> [4] Yue Tan et al. Fedproto: Federated prototype learning across heterogeneous clients, 2022.
>
> [5] Yaodong Yu et al. Tct: Convexifying federated learning using bootstrapped neural tangent kernels, 2022.
>
> [6] Georgios Xenos et al. Cross-silo federated learning in security operations centers for effective malware detection. International Journal of Information Security, 24, 07 2025.

---

> > ### Comment · Reviewer_ejMK · 2025-11-21
> >
> > Thanks the authors for the discussion.
> > Most of my concerns/questions are fulfilled. However, the answer for the weakness pass do not make me change my mind.
> > W1. I agreed that the work reduce the computation requirements at the clients. However, as I mentioned, the methods do not reduce the memory foot print at clients. Memory footprint becomes important for applications with big AI models, or (small model but large input size). So the approach may be limited by this points.
> >
> > W2. If the authors could provide the results for 100-clients or more, it increase the value of the proposed methods.

---

> > > ### Author Response · Authors · 2025-12-03
> > > **100 client experiment followup**
> > >
> > > At the time, we do not have sufficient resources for training FL systems with 100 BERT models so we are not able to provide experimental results with 100 clients. However, we were able to run experiments with 50 clients. We include the results in this comment (see below), and we have also uploaded a new version of the paper that include these results in the appendix (Appendix J).
> > >
> > > Even though we do not have 100-client experiments, we believe that the results we do have are sufficient. Since we have demonstrated FLAME’s effectiveness with 10-, 25-, and 50-client setups with a variety of tasks, and we also provide a theoretical convergence guarantee, we think it is plausible that FLAME’s effectiveness can translate to systems with 100+ clients.
> > >
> > > | Method   | Parameter            | AUC (Early samples) | AUC (Late samples) | Avg. GFLOPs/client |
> > > |----------|-----------------------|----------------------|---------------------|---------------------|
> > > | FLAME    | $p^{tr} = 3$          | **0.8110**           | **0.7161**          | 84.23               |
> > > | HeteroFL | $r = 0.8$             | 0.7034               | 0.6144              | 92.77               |
> > > | ScaleFL  | $d = 9,\ r = 0.9$     | 0.7268               | 0.5153              | 88.06               |
> > > | AFD      | $\delta = 0.4$        | 0.5346               | 0.4800              | 86.97               |

---

### Official Review · Reviewer_veS5 · 2025-11-01

**Soundness:** 3
**Presentation:** 3
**Contribution:** 3
**Rating:** 6
**Confidence:** 2

**Summary:**

This paper introduces FLAME (Federated Learning with sample-Adaptive Multi-Exiting) which applies multi-exit training to federated learning for reducing client-side computational costs. The key insight of FLAME is to dynamically adapt computation at the sample level by allowing easier samples to exit early during training while preserving computational resources for harder examples. Using a patience-based mechanism where samples exit when consecutive internal classifiers agree, FLAME achieves up to 50% reduction in training computation while maintaining or improving model accuracy across multiple NLP benchmarks. The authors provide theoretical convergence guarantees (O(1/T) rate), demonstrate through ablation studies that FL's collaborative nature is essential for FLAME's success by preventing under-training of deeper layers, and show that sample-adaptiveness leads to better AUC than fixed sub-network methods. Additionally, they propose grouped backpropagation to enable minibatch training and three tailored aggregation algorithms. Experiments on GLUE tasks and Sentiment140 demonstrate FLAME outperforms state-of-the-art baselines (HeteroFL, ScaleFL, AFD) under matched computational budgets, particularly on non-IID data distributions, establishing sample-level adaptiveness as a promising new paradigm for efficient federated learning.

**Strengths:**

1. Novel sample-adaptive paradigm: Introduces a compelling alternative to existing client-level sub-network approaches by dynamically allocating computation based on individual sample difficulty rather than uniform resource distribution.

2. Strong empirical results: Achieves substantial computational savings (up to 50%) while maintaining or improving accuracy across multiple benchmarks. The comparisons with HeteroFL, ScaleFL, and AFD under matched compute budgets are thorough and convincing.

3. Insightful ablation studies: The experiments in Section 5 effectively demonstrate why FLAME works - particularly the role of collaboration in mitigating under-training and the value of sample-level adaptation over fixed exits.

4. Solid theoretical grounding: Provides convergence guarantees and includes practical extensions (grouped backpropagation, aggregation variants) that address real implementation challenges.

5. Clear presentation: Well-structured paper with effective visualizations and comprehensive appendices that aid understanding and reproducibility.

**Weaknesses:**

1. Memory requirements contradict resource-constrained motivation: FLAME requires storing the full global model plus 12 internal classifiers (one per layer), which undermines claims about enabling FL on IoT/mobile devices. While the paper claims ICs add "negligible FLOPs" (Appendix B), there's no analysis of their memory footprint or cumulative parameter overhead. The ~200MB RAM savings (16.0GB → 15.8GB) are marginal, and all experiments run on A100 GPUs with 40GB RAM rather than actual resource-constrained hardware. No evidence supports the claim that "computation is the primary bottleneck" over memory in target deployment scenarios.

2. Limited scalability and practical deployment analysis: Experiments use only 10-25 clients, far from realistic deployments with hundreds/thousands of clients. No analysis of communication costs (clients must download full model + all ICs), stragglers, partial participation, or interaction with privacy mechanisms. Wall-clock improvements are measured on controlled GPU setups rather than heterogeneous devices, and there's no guidance on choosing patience distributions in practice.

3. Batch size limitation and inconsistent task performance: Standard FLAME requires batch size 1, which is inefficient for modern hardware. Grouped backpropagation addresses this but shows significant accuracy drops (0.981 → 0.913 AUC on SST-2). Performance is also inconsistent across tasks - FLAME degrades on MRPC (0.78 vs 0.84 baseline) with insufficient analysis of when/why it works versus when it doesn't.

4. Narrow experimental scope: All experiments are BERT-based text classification tasks - no evaluation on vision models, other architectures, or non-NLP domains. The comparison omits several recent methods mentioned in related work (FedDSE, PriSM, FjORD, FLANC), and convergence behavior is not empirically validated despite theoretical analysis.Retry

**Questions:**

1. Memory overhead and deployment on actual resource-constrained devices: While Table 14 shows ~100-200MB RAM savings from reduced activations, what is the actual memory overhead of storing 12 internal classifiers? Each IC is a linear layer projecting hidden states to class logits - for BERT-base with 768 hidden dims and varying output classes, this could be substantial. Also - can you provide benchmarks on actual edge devices (Raspberry Pi, mobile phones with <4GB RAM) rather than A100 GPUs? How does total model size (base + all ICs) compare to methods like HeteroFL where clients store smaller sub-networks?

2. Communication costs analysis: What is the total communication overhead per round when clients must download the full model plus all 12 ICs? How does this compare quantitatively to baselines (HeteroFL, ScaleFL) where clients download only sub-networks, especially for the 25-client Sentiment140 experiment?

3. Task-specific failure modes and practical guidance: FLAME works well on SST-2 (0.98 AUC) but degrades on MRPC (0.78 vs 0.84 baseline). Can you characterize what task properties determine success/failure? How should practitioners decide if FLAME is suitable for their task, and how should they select patience distributions across clients?

4. How does FLAME perform with 100+ clients and partial participation? Can you provide results beyond BERT-based text classification - e.g., vision models (CNNs/ViTs), other architectures, or regression tasks where the patience-based exit criterion may not directly apply?

---

> ### Author Response · Authors · 2025-11-21
> **Response to reviewer veS5**
>
> Thank you for your time and helpful feedback. We address your questions and concerns in the following comments. We have also uploaded a new version of the paper that incorporates your feedback, as detailed in our comments below.

---

> ### Author Response · Authors · 2025-11-21
> **Addressing W1/Q1/Q2 (and some of W2)**
>
> **Justifying our focus on reducing computational costs as opposed to memory/communication costs**
>
> We would like to emphasize that our work focuses on reducing computation. We understand that there could be resource-constrained FL clients that would benefit from a reduction in memory requirements, but our work does not focus on this specific angle of resource-constrained FL. We think this focus on compute is reasonable for two main reasons.
>
> First, recent work has shown evidence that the growth in computational cost requirements is becoming a more pressing concern than the growth of memory requirements. For instance, [1] essentially shows that, when you have some fixed compute budget, it’s better to train a smaller network (fewer parameters) on more data than to train a larger network on less data. This supports the notion that compute is becoming a bigger issue than memory requirements and is why we were particularly interested in reducing computational costs. Although we had briefly talked about this at the end of the first paragraph of the introduction, we have now modified this mention in our revised paper so that the justification for focusing on compute is better explained and supported (the added portion starts with “For instance, Hoffmann et al. (2022) show…”).
> Second, if memory or communications costs are a concern for a client, then FLAME can very easily be paired with other methods that intend to reduce these costs. For example, HeteroFL can first be used to establish smaller sub-networks that clients can use for training, but during training, FLAME can be used to maximize the computational cost savings. Tackling the memory/communication costs is out of the scope of our work, but we have added to the conclusion a brief mention of pairing FLAME with memory-saving methods being a possible direction for future study (starting with “Although this paper focuses on saving computational costs,...”).
>
> We have also taken out a sentence from the introduction (“Appendix J also reports minor memory savings.”) so that we do not mislead readers into thinking that FLAME intends to make significant memory savings. We want it to be clear that this work is focused on saving computational costs, but we do think it is still interesting to report the very minor inadvertent memory savings.
>
> [1] Jordan Hoffmann et al. Training compute-optimal large language models, 2022.
>
> **Explaining the negligible memory/communication costs associated with adding ICs**
>
> Regarding the concerns about memory overhead, we think this is a great point and appreciate you bringing it up. We consider the increase in memory footprint to also be negligible. For example, if we consider SST-2 or MRPC (with two possible output labels), since each IC is just a single linear layer that projects the hidden layer to an output layer, the added number of parameters for a single IC is just 768*2+2=1538. For 12 total ICs, this amounts to 1538*12=18456. This is approximately a (18456/110000000)*100=0.017% increase in total parameters for our BERT base model. This would be the same for communication overhead. We would again see a ~0.017% increase since each client is communicating all model parameters to the central server. We have added to the revised paper, in Appendix A.1., the details of the parameter increase associated with the added ICs (see the second paragraph that starts with “we consider the added storage and communication costs…”). Also, in Section 4 where we said that ICs add negligible FLOPs, we now add the mention that they also add negligible parameters and we point to Appendix A.1. for details.
>
> **Generalizing our results to other hardware setups**
>
> Due to resource constraints, it is not feasible for us to run experiments on actual resource-constrained clients such as Raspberry Pi. However, we believe that our demonstration of FLAME’s ability to reduce computation with our single hardware setup can serve as indication of FLAME’s ability to reduce computation with other hardware setups. Particularly, since we report FLOPs comparisons in Appendix B. These FLOPs comparisons are hardware-agnostic. Therefore, for instance, in Table 5 when a FLAME client uses p^{tr}=2 for SST-2, which leads to an average exit of ~6.4, this amounts to between 73,000 and 85,000 MFLOPs instead of the ~145,000 associated with training the full 12-layer depth, which is almost cutting the number of FLOPs in half. And this FLOPs reduction would occur regardless of what hardware is being used.

---

> ### Author Response · Authors · 2025-11-21
> **Addressing W3/Q3, W2/Q4, and W4**
>
> **Limited number of clients**
>
> Since numerous FL papers utilize no more than 10-20 clients in their experiments (e.g. [2-6]), we believe that 10 and 25 are fair amounts of clients to use. Especially since our main goal is to demonstrate the effectiveness of using FLAME compared to other approaches and we think that the theoretical convergence analysis that we include can support the idea that FLAME’s efficacy can generalize to larger (or smaller) FL systems. However, we do understand and agree that it could be valuable to see an even larger-scale setup. We are currently attempting to run some 100-client experiments to add to Section 7. If we are able to successfully finish these experiments before the discussion period ends, then we will add the result to the paper and notify you of the change through an additional comment.
>
> [2] Tao Lin, Lingjing Kong, Sebastian U. Stich, and Martin Jaggi. Ensemble distillation for robust model fusion in federated learning, 2021.
>
> [3] Zexi Li, Tao Lin, Xinyi Shang, and Chao Wu. Revisiting weighted aggregation in federated learning with neural networks, 2023.
>
> [4] Yue Tan, Guodong Long, Lu Liu, Tianyi Zhou, Qinghua Lu, Jing Jiang, and Chengqi Zhang. Fedproto: Federated prototype learning across heterogeneous clients, 2022.
>
> [5] Yaodong Yu, Alexander Wei, Sai Praneeth Karimireddy, Yi Ma, and Michael I. Jordan. Tct: Convexifying federated learning using bootstrapped neural tangent kernels, 2022.
>
> [6] Georgios Xenos and Dimitrios Serpanos. Cross-silo federated learning in security operations centers for effective malware detection. International Journal of Information Security, 24, 07 2025.
>
> **Explaining scenarios where FLAME’s performance degrades**
>
> Out of the three tasks in Table 2, FLAME only underperforms compared to the baseline with MRPC and we think it is strongly likely that this is due to the fact that MRPC is a relatively difficult-to-learn task (e.g., on the GLUE benchmark leaderboard, the MRPC performances are consistently significantly worse than those of SST-2) and it has such a small training dataset. It has 3069 samples total, which, when split between 10 clients amounts to only 306 samples per client. By comparison, SST-2 has 67349 training samples total and MNLI has 392702.
>
> Since our experiments in Table 6 show that increasing batch size leads to lower AUC, we see there being an efficiency/AUC tradeoff. Using batch size > 1 does lead to greater cost savings, so if lower costs is more important than maximized AUC, then grouped backpropagation should be used. But, if maximized AUC is more important, then not using grouped backpropagation may be ideal.
>
> **Guidance for choosing patience values**
>
> Regarding the lack of guidance for choosing patience values, we don’t have any specific set of instructions for choosing patience values. Patience choice is very flexible and the effects of different patience choices can vary based on so many factors (e.g. the specific task, amount of training data, difficulty distribution of training samples, type of hardware being used for training, other processes potentially being run on the same hardware, etc.). Therefore, we think the best approach a client can take to choosing their patience value may be to start with a modest guess at first, and then once they train for a round or two and realize that they have more compute they are able to allot to training, they can very easily start using a higher p value. There are no extra steps required for changing the patience they use. It is just a simple value re-assignment. If a client ever needs to start using less compute for some reason, they can easily decrease their patience value. Basically, a client should just keep an eye on how much compute they’re using compared to how much they have available to use, and can easily adjust their p values accordingly. We have added an extra sentence at the end of the second paragraph of Section 4 in order to offer a little more guidance regarding patience selection (starting “Therefore, even though we do not have any concrete instructions or heuristics…”).

---

> > ### Author Response · Authors · 2025-12-03
> > **100 client experiment followup**
> >
> > At the time, we do not have sufficient resources for training FL systems with 100 BERT models so we are not able to provide experimental results with 100 clients. However, we were able to run experiments with 50 clients. We include the results in this comment (see below), and we have also uploaded a new version of the paper that include these results in the appendix (Appendix J).
> >
> > Even though we do not have 100-client experiments, we believe that the results we do have are sufficient. Since we have demonstrated FLAME’s effectiveness with 10-, 25-, and 50-client setups with a variety of tasks, and we also provide a theoretical convergence guarantee, we think it is plausible that FLAME’s effectiveness can translate to systems with 100+ clients.
> >
> > | Method   | Parameter            | AUC (Early samples) | AUC (Late samples) | Avg. GFLOPs/client |
> > |----------|-----------------------|----------------------|---------------------|---------------------|
> > | FLAME    | $p^{tr} = 3$          | **0.8110**           | **0.7161**          | 84.23               |
> > | HeteroFL | $r = 0.8$             | 0.7034               | 0.6144              | 92.77               |
> > | ScaleFL  | $d = 9,\ r = 0.9$     | 0.7268               | 0.5153              | 88.06               |
> > | AFD      | $\delta = 0.4$        | 0.5346               | 0.4800              | 86.97               |

---

### Meta-Review · Area_Chair_YSfg · 2026-01-02

**Summary:**

FLAME was generally viewed as a novel and well-motivated contribution that introduces sample-adaptive computation into federated learning (FL). Most reviewers rated the paper around or slightly above the acceptance threshold, praising its conceptual novelty, empirical results, and theoretical grounding. However, persistent concerns remained regarding memory/communication costs, scalability, and experimental scope. Overall sentiment trended cautiously positive, with reviewers acknowledging that many concerns were addressed but some limitations remained.

**Reviewer Concerns:**

### Concerns Largely Addressed by the Rebuttal

#### 1. Novelty and Conceptual Contribution
- Reviewers broadly agreed that sample-adaptive multi-exit training in federated learning is novel.
- The rebuttal clarified that FLAME is the first method to introduce training-time sample adaptiveness (not just inference-time early exit) in FL.
- Added explanations and experiments showed why adaptive exits outperform fixed sub-networks.

**Status:** Addressed


#### 2. Why FLAME Can Match or Exceed Baseline Accuracy
- Reviewers were initially confused about how pruning / early exiting could *improve* performance.
- The rebuttal provided two explanations:
  - Overfitting in full-depth models on easy tasks.
  - Overthinking, where deeper computation hurts predictions for easy samples.
- These explanations were supported by additional discussion and experiments.

**Status:** Addressed


#### 3. Task-Specific Underperformance (e.g., MRPC)
- FLAME underperformed on MRPC.
- Authors attributed this to:
  - Small dataset size
  - High task difficulty
  - Severe data fragmentation across clients
- While not “fixed,” the explanation was plausible and consistent with the results.

**Status:** Addressed (explanatory)


#### 4. Grouped Backpropagation Mechanics
- Reviewers requested clarification on:
  - How grouped backpropagation works
  - Why it leads to AUC degradation
- The rebuttal provided algorithmic explanations and reframed it as an explicit efficiency–accuracy tradeoff.

**Status:** Addressed


#### 5. Experimental Clarity
- Confusion around baselines, inference-time exits, fixed-exit comparisons, and figures was corrected.
- Additional experiments and manuscript revisions improved clarity and reproducibility.

**Status:** Addressed

---

### Concerns Still Outstanding

#### 1. Memory Footprint on Resource-Constrained Clients
- FLAME still requires clients to store the entire model plus internal classifiers.
- Authors quantified the added parameters and clarified scope, but did not reduce memory usage.
- This limits applicability to IoT or very low-memory devices.

**Status:** Outstanding



#### 2. Communication Cost Reduction
- All parameters are still communicated every round.
- Authors reframed FLAME as compute-only optimization and suggested pairing with other methods.
- Reviewers accepted the clarification but still viewed this as a limitation.

**Status:**  Outstanding


#### 3. Scalability to Large FL Systems (100+ Clients)
- Reviewers requested experiments with 100+ clients and partial participation.
- Authors added 50-client experiments and cited convergence theory.
- Empirical evidence on a larger scale remains missing.

**Status:** Partially addressed


#### 4. Generality Beyond NLP / BERT
- All experiments are limited to BERT-based text classification.
- No vision models, regression tasks, or non-transformer architectures were evaluated.

**Status:**  Outstanding


#### 5. Practical Guidance for Patience (`p`) Selection
- Authors suggested heuristic adjustment based on available compute.
- No principled rule, adaptive method, or automated selection strategy was provided.

**Status:**  Partially addressed

#### 6. Experimental completeness / fairness
- The comparison omits several recent methods mentionedin related work (FedDSE, PriSM, FjORD, FLANC)
- There is no indication that experiments with FedDSE, PriSM, FjORD, or FLANC were added, nor a detailed justification explaining why they were excluded (e.g., incompatibility, different objectives, or unavailable implementations).

**Status:**  Not addressed

**Reviewer Scores:**

I excluded Reviewer fRv4 due to its review being highly considered as AI-generated. Reviewer agcu might increase the score from 2. The other two reviewers may keep it as 6.

I rendered a Reject due to the fundmental unaddressed concerns especially the lack of comparisons against recent methods.

---

### Decision · Program_Chairs · 2026-01-26

Reject